# Effector CD4$^+$ T cells recognize intravascular antigen presented by patrolling monocytes

Clare L.V. Westhorpe[1], M. Ursula Norman[1], Pam Hall[1], Sarah L. Snelgrove[1], Michaela Finsterbusch [1,6], Anqi Li[1], Camden Lo[2], Zhe Hao Tan[1], Songhui Li[3,4], Susan K. Nilsson [3,4], A. Richard Kitching [1,5] & Michael J. Hickey [1]

Although effector CD4$^+$ T cells readily respond to antigen outside the vasculature, how they respond to intravascular antigens is unknown. Here we show the process of intravascular antigen recognition using intravital multiphoton microscopy of glomeruli. CD4$^+$ T cells undergo intravascular migration within uninflamed glomeruli. Similarly, while MHCII is not expressed by intrinsic glomerular cells, intravascular MHCII-expressing immune cells patrol glomerular capillaries, interacting with CD4$^+$ T cells. Following intravascular deposition of antigen in glomeruli, effector CD4$^+$ T-cell responses, including NFAT1 nuclear translocation and decreased migration, are consistent with antigen recognition. Of the MHCII$^+$ immune cells adherent in glomerular capillaries, only monocytes are retained for prolonged durations. These cells can also induce T-cell proliferation in vitro. Moreover, monocyte depletion reduces CD4$^+$ T-cell-dependent glomerular inflammation. These findings indicate that MHCII$^+$ monocytes patrolling the glomerular microvasculature can present intravascular antigen to CD4$^+$ T cells within glomerular capillaries, leading to antigen-dependent inflammation.

[1] Centre for Inflammatory Diseases, Monash University Department of Medicine Monash Medical Centre, 246 Clayton Rd., Clayton, VIC 3168, Australia. [2] Monash Micro Imaging, Monash University, Wellington Rd., Clayton, VIC 3800, Australia. [3] Biomedical Manufacturing, CSIRO Manufacturing, Bag 10, Clayton South, VIC 3169, Australia. [4] Australian Regenerative Medicine Institute, Monash University, Wellington Rd., Clayton, VIC 3800, Australia. [5] Departments of Nephrology and Pediatric Nephrology, Monash Medical Centre, 246 Clayton Rd., Clayton, VIC 3168, Australia. [6] Present address: Department of Vascular Biology and Thrombosis Research, Medical University of Vienna, Schwarzspanierstr. 17, 1090 Vienna, Austria. Clare L.V. Westhorpe and M. Ursula Norman contributed equally to this work. Correspondence and requests for materials should be addressed to M.J.H. (email: michael.hickey@monash.edu)

A growing body of evidence indicates that immune cells can make critical contributions to inflammatory responses while remaining within the vasculature[1]. This concept of "intravascular immunity" is exemplified by the intravascular migration of non-classical monocytes in tissues such as skin, mesentery, muscle, and brain[2–6]. In vivo imaging studies show that this patrolling function involves prolonged crawling on the endothelium independent of the direction of blood flow[2,4]. Patrolling Ly6C$^-$ monocytes perform important immune surveillance within the vasculature, internalizing microparticles and soluble material from the bloodstream and responding to microbial infection or tissue injury[3,7]. Upon detection of these signals, intravascular monocytes are positioned to respond rapidly by inducing recruitment of other immune cells or migrating out of the vasculature[3–5]. These intravascular activities are not restricted to myeloid leukocytes as in the liver microvasculature, invariant natural killer T (iNKT) cells also constitutively migrate. In this location iNKT cells respond to innate and adaptive signals by modulating their migration and releasing proinflammatory mediators[8,9]. Whether these intravascular functions also contribute to adaptive immune responses involving conventional T cells is less clear.

Intravital imaging studies have revealed that CD4$^+$ and CD8$^+$ T cells in microvessels of the central nervous system and the renal interstitium can undergo intraluminal crawling and subsequently exit the vasculature[10–12]. While in contact with the endothelium, effector CD8$^+$ cells can recognize peptide/MHC class I (MHCI) complexes expressed by endothelial cells, leading to T-cell activation and promotion of recruitment and/or tissue retention[12–14]. Although some endothelial cells can express MHC class II (MHCII) in inflammatory states, mechanisms of antigen recognition by antigen-specific CD4$^+$ T cells within the vasculature are unclear[15–17]

One site where intravascular immunity is crucial is the specialized microvasculature of the glomerulus. Monocytes and neutrophils have been shown to undergo constitutive intravascular adhesion and crawling within glomerular capillaries[17,18]. During antibody-mediated glomerular inflammation, the duration of the retention of these cells is increased and intravascular neutrophils generate reactive oxygen species (ROS) responsible for glomerular injury[17,18]. In addition to humoral mediators, CD4$^+$ T cells also have an important function in the development of glomerular injury and dysfunction in severe, rapidly progressive forms of glomerulonephritis. Intraglomerular T cells are detectable in humans with rapidly progressive glomerulonephritis[19–21], and functional studies in animal models demonstrate that disease-inducing effector responses can be directed by CD4$^+$ T cells responding to antigen located intravascularly within the glomerulus[15,16,22–24].

Evidence indicates that glomerular injury mediated by effector CD4$^+$ T cells in glomerulonephritis involves multiple steps, beginning with loss of tolerance to nephritogenic autoantigens in secondary lymphoid organs[16,25]. This evidence includes the discovery of circulating CD4$^+$ T cells specific for these antigens in patients with autoimmune glomerulonephritis[26–28]. In patients with autoimmune disease, circulating autoreactive T cells have a memory phenotype, indicating that they have been exposed to cognate antigen and undergone differentiation into effector or memory T cells[29,30]. However, the existence of circulating, antigen-experienced T cells is insufficient to result in disease. The final steps in the process involve T-cell recognition of antigen in the target tissue, leading to effector T-cell-mediated injury at the site of antigen recognition, the glomerulus. Indeed, analysis of antigen-experienced T-cell responses to antigen in the periphery has shown that these cells can respond within minutes upon recognition of cognate antigen[31,32]. However, the mechanism

whereby CD4$^+$ T cells recognize antigens in the unique microvasculature of the glomerulus is not known. Therefore, the aim of this study is to investigate the mechanisms of intravascular antigen presentation to disease-initiating, antigen-experienced effector CD4$^+$ T cells in the glomerulus, using a validated model of T-cell-mediated glomerulonephritis. The findings indicate that in the absence of inflammation, MHCII expression in the glomerulus is restricted to subsets of circulating leukocytes. Of these cells, monocytes undergo the most prolonged retention and migration in the glomerular capillaries and are required for CD4$^+$ T-cell-mediated induction of neutrophil-dependent glomerular inflammation.

## Results

**CD4$^+$ T cells migrate in uninflamed glomerular capillaries.** We first examined whether CD4$^+$ T cells could spontaneously adhere within uninflamed glomerular capillaries. Endogenous CD4$^+$ T cells were visualized using anti-CD4 mAb and intravital multiphoton imaging of the kidney. In uninflamed glomeruli, CD4$^+$ T cells regularly underwent periods of adhesion (defined as retention for >30 s) on the endothelial surface, at a rate of ~5 cells per glomerulus per hour (Fig. 1a, b and Supplementary Movie 1). The majority remained stationary during adhesion ("static" cells), while ~20% underwent crawling (Fig. 1b). Typically, the duration of retention, or dwell time, of CD4$^+$ T cells was ~4 min (Fig. 1c). Crawling CD4$^+$ T cells were retained in glomeruli for twice as long as stationary cells and migrated at ~9 µm min$^{-1}$ (Fig. 1c, d).

Detection of these blood-borne CD4$^+$ T cells in this fashion does not allow the differentiation between naïve and effector T cells. This is important as for CD4$^+$ T cells to induce disease in the glomerulus in response to local antigen recognition, they require prior activation and differentiation into effector cells in secondary lymphoid organs. Therefore, we next asked whether activated effector T cells also undergo retention in uninflamed glomerular capillaries. To examine this issue, we used OVA-specific T cells from TCR transgenic OT-II mice. OT-II T cells were activated in vitro to a Th1 phenotype, fluorescently labeled and transferred intravenously into recipient mice and subsequently glomeruli were examined via intravital multiphoton imaging. As for endogenous CD4$^+$ T cells, activated OT-II cells underwent retention in glomerular capillaries (Fig. 1e, Supplementary Movie 2). On average, ~1.5 OT-II cells underwent adhesion per glomerulus per hour, of which ~70% remained stationary (Fig. 1f). OT-II cells arrested in glomeruli at a relatively consistent rate during the 90 min imaging period after transfer (Fig. 1g). The dwell time of OT-II cells in glomeruli was ~13 min —this was similar in stationary and crawling OT-II T cells (Fig. 1h) while more than twice that of endogenous CD4$^+$ T cells. The mean migration velocity of crawling OT-II cells was ~9 µm min$^{-1}$ (Fig. 1i). Together these findings indicate that CD4$^+$ T cells, and specifically effector CD4$^+$ T cells, undergo adhesion and migration in the glomerular microvasculature in the absence of inflammation.

**T-cell adhesion in antigen-bearing glomerular capillaries.** We have previously shown that OT-II cells can trigger antigen-dependent neutrophil recruitment within 4 h in a planted antigen model of CD4$^+$ T-cell-dependent glomerulonephritis[15–17]. In this model, the peptide antigen OVA$_{323–339}$ (pOVA) is delivered to the glomerular microvasculature via conjugation to 8D1, an mAb that binds to the NC1 domain of the α3 chain of type IV collagen in the glomerular basement membrane without inducing glomerular injury[33]. The 8D1/pOVA construct was transferred into mice intravenously together with fluorescently labeled OT-II cells. In order to focus on initiation of the T-cell-dependent response,

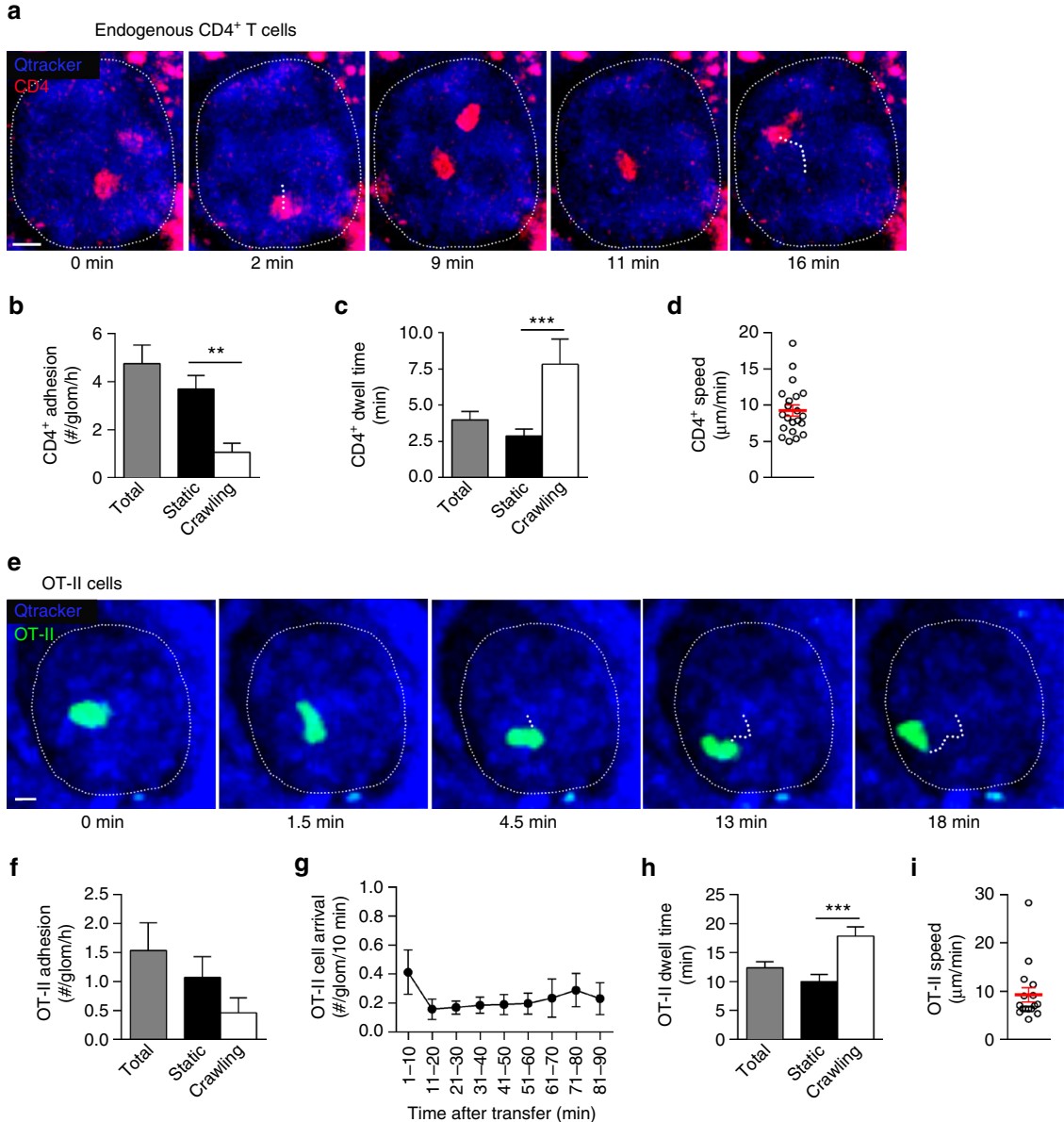

**Fig. 1** CD4$^+$ T cells migrate constitutively in the uninflamed glomerular microvasculature. **a–d** Retention and migration of endogenous CD4$^+$ T cells in the glomerular microvasculature of untreated mice, as assessed using multiphoton intravital microscopy. **a** Image sequence showing CD4$^+$ T cells (anti-CD4-PE, red) undergoing retention and migration in the glomerular capillaries (vasculature detected via Qtracker® 655—blue). The glomerular border is denoted by a thin dotted line, and the migration paths of two CD4$^+$ T cells are indicated by thick dotted lines (time elapsed shown below images). See also Supplementary Movie 1. Scale bar, 10 μm. **b–d** Quantification of adhesion and migration of endogenous CD4$^+$ T cells in glomeruli of untreated mice ($n = 6$ mice). Data show number (**b**) and dwell time (**c**) of adherent CD4$^+$ T cells, and velocity of crawling cells (**d**). In **b** and **c**, data are shown for total cells, and specifically for static or crawling cells. In **b**, data are expressed as # per glomerulus per h per mouse. In **c**, data are expressed per cell ($n = 124$ total, 96 static and 28 crawling). In **d**, circles represent individual cells ($n = 21$). **e–i** Retention and migration of effector CD4$^+$ T cells in the glomerular microvasculature of untreated mice. OT-II cells were activated in vitro, labeled with CFSE (green) and transferred into uninflamed mice. **e** Image sequence showing effector CD4$^+$ T cell (green) migrating within the glomerular capillaries. See also Supplementary Movie 2. Scale bar, 10 μm. **f–i** Quantification of adhesion and migration of activated OT-II cells in glomeruli in the 90 min following transfer ($n = 8$ mice). **f** Number of adherent cells, shown for total, static, and crawling cells, expressed as OT-II cells per glomerulus per h per mouse. **g** Rate of OT-II cell adhesion, assessed in 10 min intervals. **h** Dwell time of total, static, and crawling OT-II cells ($n = 333$ total, 232 static and 101 crawling). **i** Migration speed of crawling OT-II cells. In **i**, circles represent individual cells ($n = 16$). In **b–d** and **f–i**, data are shown as mean ± s.e.m. (as red lines in **d** and **i**). **$P < 0.01$; ***$P < 0.001$. **b, c, f** and **h**: unpaired Student's $t$-tests

glomeruli were imaged for 2 h after OT-II transfer. Compared to control mice receiving unconjugated 8D1 mAb, more OT-II cells adhered in glomerular capillaries of mice receiving 8D1/pOVA (Fig. 2a), due solely to an increase in the number of crawling cells (Fig. 2b). OT-II cells also crawled more slowly in mice that received 8D1/pOVA (Fig. 2c), a finding consistent with the reduced velocity of effector T cells during antigen recognition

described in other organs[31,32,34]. To identify when the increased retention of crawling OT-II cells occurred, the rate of OT-II cell recruitment to the glomerular capillaries was examined in 10 min intervals during the 2 h after transfer. In mice given 8D1/pOVA, a significant increase in crawling OT-II cells was detectable 71–80 min after transfer (Fig. 2d). In contrast, the rate of arrest of static OT-II cells was constant after the first 20 min of the observation

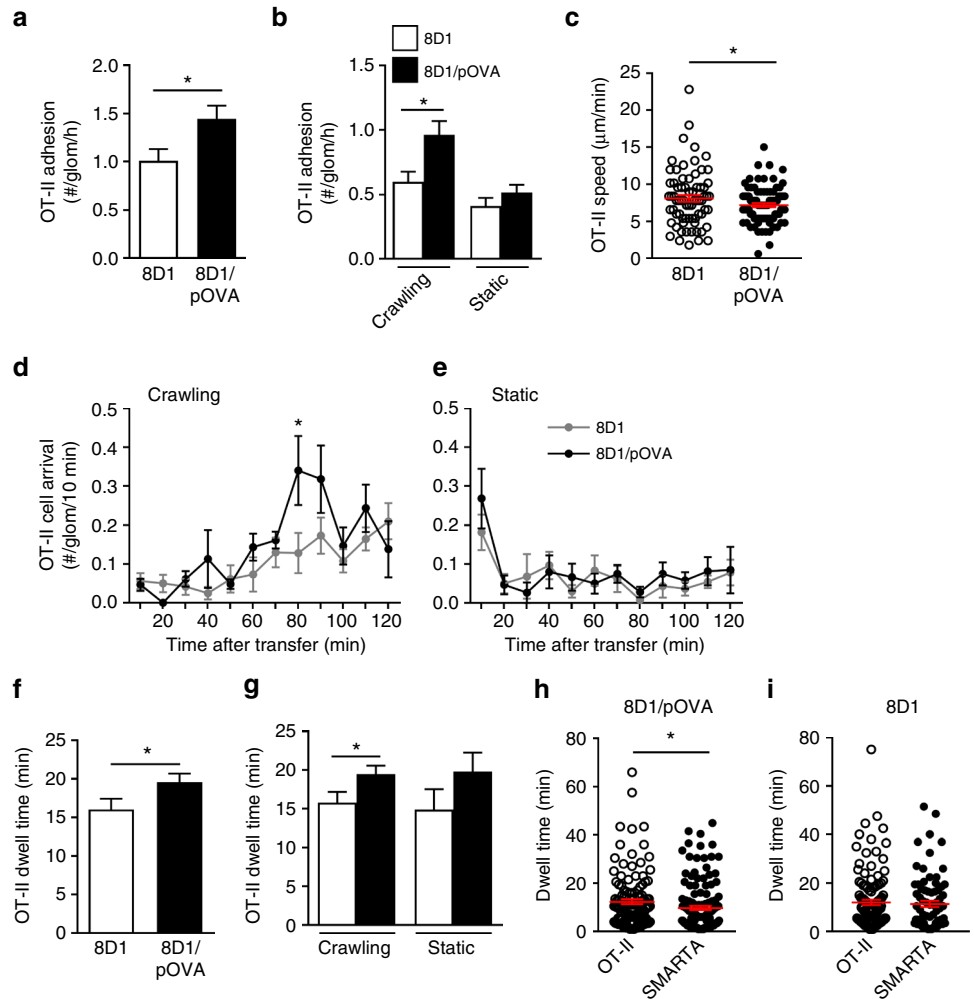

**Fig. 2** Presence of antigen in glomeruli alters recruitment and migration of effector CD4+ T cells. Retention and migration of activated OT-II cells in glomerular capillaries were assessed via intravital multiphoton microscopy after i.v. administration of OT-II cells and either unconjugated control 8D1 mAb or 8D1/pOVA ($n = 6$ mice per group). Data are shown for the total number of adherent OT-II cells (**a**), the number of crawling or static OT-II cells (**b**) and the crawling speed of OT-II cells (**c**) (8D1, $n = 71$ cells; 8D1/pOVA $n = 83$ cells). **d, e** Rate of arrival of adherent OT-II cells as assessed in 10-min intervals in the 2 h following administration, for crawling cells (**d**) and static cells (**e**). **f, g** Dwell time for OT-II cells, shown for total cells (8D1, $n = 226$; 8D1/pOVA, $n = 423$) (**f**), and for crawling (8D1, $n = 132$; 8D1/pOVA, $n = 257$) and static (8D1, $n = 90$; 8D1/pOVA, $n = 148$) cells (**g**). **h, i** Activated OT-II (pOVA-specific) and SMARTA (LCMV-specific) CD4+ T cells were co-transferred into mice treated with either 8D1-pOVA (**h**) or 8D1 (**i**), and glomerular retention of the two types of cells was assessed using multiphoton microscopy. Shown are individual T-cell dwell times 30–120 min after cell transfer, as well as group mean ± s.e.m. Data represent analysis of a total 139 SMARTA and 127 OT-II cells (**h**) or 90 SMARTA and 125 OT-II cells (**i**) from $n = 3$ recipient mice in both experiments. *$P < 0.05$ for the comparisons shown. **a**, **b** Mann−Whitney tests; **c**, **d**, **f**, **g**, **h**, **i** unpaired Student's t-tests

period and did not differ between the groups (Fig. 2e). OT-II cells were also retained in glomerular capillaries for longer periods in mice that received 8D1/pOVA (Fig. 2f), due to a significant increase in the dwell time of crawling OT-II cells (Fig. 2g). To control for possible effects of inflammation-associated alterations in glomerular hemodynamics on the retention of OT-II cells[35], we also compared retention of OT-II T cells with that of T cells from SMARTA mice, which recognize an unrelated peptide. In co-transfer experiments with differentially labeled activated OT-II and SMARTA cells, in mice treated with 8D1-pOVA, OT-II cells showed prolonged glomerular dwell time relative to co-transferred SMARTA T cells (Fig. 2h). This response was not seen in the absence of antigen (Fig. 2i), supporting the hypothesis that the increased retention of OT-II cells was due to their capacity to recognize pOVA, rather than physical factors at play in the inflamed glomerulus.

We also examined intravascular migration by OT-II cells in glomerular capillaries 2–3 h after cell transfer, when the number of adherent OT-II cells was comparable with or without exposure to antigen and similar for crawling and static cells (Supplementary Fig. 1a, b). However, consistent with the earlier time point (Fig. 2g), in mice receiving antigen, crawling effector OT-II cells remained in glomeruli for longer, and crawled more slowly (Supplementary Fig. 1c, d) during the 2–3 h period post-transfer. These results indicate that effector OT-II cells undergo increased retention in glomerular capillaries in response to a planted glomerular antigen within just over an hour of administration.

**Antigen-presenting cells patrol glomerular capillaries**. Effector CD4+ T cells recognize antigenic peptides presented via MHCII molecules to induce responses in peripheral tissues. However, in non-inflamed glomeruli, it has been reported that resident cells expressing MHCII are rare[36]. We confirmed these findings using MHCII-EGFP mice in which GFP is expressed fused to MHCII, thereby labeling MHCII-expressing cells[37]. Three-dimensional

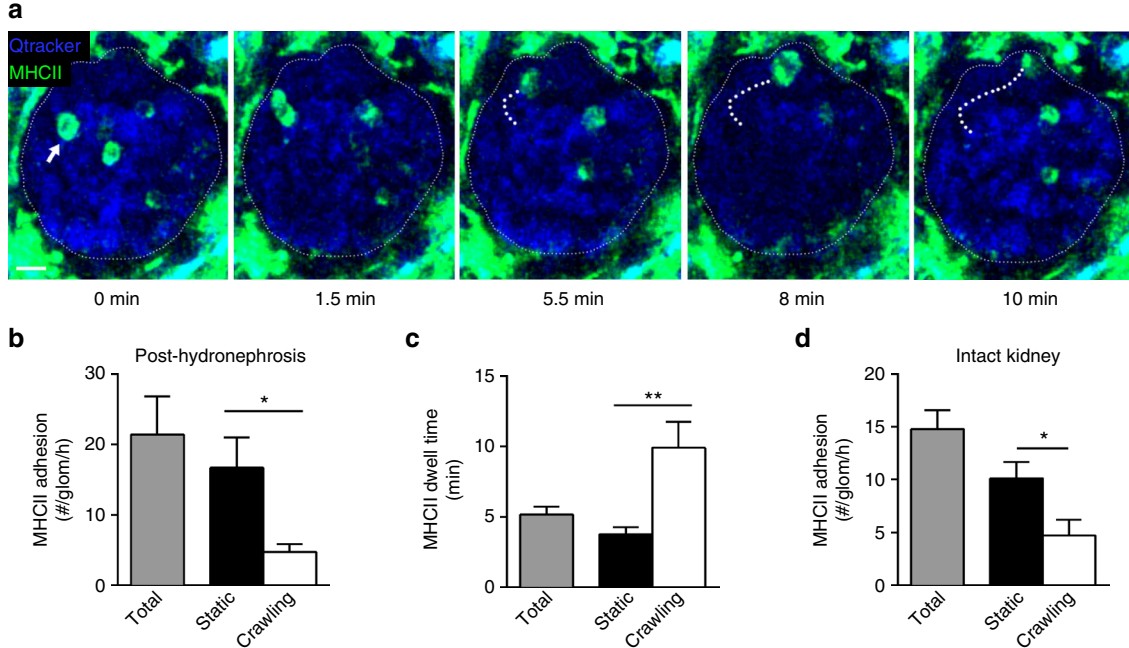

**Fig. 3** MHCII$^+$ immune cells constitutively migrate in glomerular capillaries. Multiphoton microscopy was performed on glomeruli of uninflamed MHCII-EGFP mice. **a** Image sequence showing actions of adherent EGFP$^+$ leukocytes (green) in glomerular capillaries (vasculature detected via Qtracker® 655, blue). Arrow (left panel) indicates a crawling EGFP$^+$ leukocyte. See also Supplementary Movie 5. The migration path of the cell is indicated by the thick dotted line in subsequent images (time elapsed shown below). Glomerular border is denoted by a thin dotted line. Scale bar represents 10 μm. **b–d** Quantification of adhesion and migration of MHCII$^+$ cells undergoing retention in glomeruli of untreated post-hydronephrotic mice ($n = 6$ mice). Data are shown for MHCII$^+$ leukocyte adhesion (**b**) and dwell time (**c**) (shown for total, static, crawling in both). Data in **c** are expressed per cell: total $n = 400$, static $n = 310$, crawling $n = 90$ cells. **d** Number of adherent MHCII-EGFP$^+$ leukocytes (total, static, crawling) in glomeruli of intact kidneys of 3–4-week-old mice ($n = 7$ mice). See also Supplementary Movie 6. Data are presented as mean ± s.e.m. *$P < 0.05$; **$P < 0.01$. **b**, **c** Mann−Whitney tests; **d** unpaired Student's $t$-test

multiphoton analysis of fixed tissues confirmed a lack of MHCII expression within uninflamed glomeruli, though many cells of dendritic morphology expressed MHCII in the interstitium and the periglomerular region (Supplementary Fig. 2 and Supplementary Movie 3). To exclude the possibility that projections from periglomerular mononuclear phagocytes extended into the glomerulus, we performed similar analyses of kidneys from CD11c-YFP mice, in which YFP is highly expressed in renal dendritic cells and cellular projections are readily detectable[38–40]. In agreement with previous studies, these experiments revealed an interdigitated network of dendritic cells in the interstitium, including immediately adjacent to, but not within glomeruli[39,41]. However, no projections from these cells were detected within glomeruli (Supplementary Fig. 2 and Supplementary Movie 4). These data indicate that mononuclear phagocytes of the renal interstitium are unlikely to contribute to intraglomerular antigen presentation under resting conditions.

We next asked whether the MHCII$^+$ cells responsible for antigen presentation were blood-borne leukocytes migrating within glomerular capillaries. Several populations of MHCII-expressing leukocytes circulate in the bloodstream, including B cells and a subset of monocytes[42]. To determine if MHCII$^+$ cells also undergo retention and migration in glomerular capillaries, we examined kidneys of MHCII-EGFP mice via intravital multiphoton imaging. Experiments in uninflamed MHCII-EGFP mice revealed that numerous intravascular MHCII$^+$ cells migrate within glomerular capillaries (Fig. 3a, Supplementary Movie 5). Approximately 20 MHCII$^+$ cells underwent adhesion in the glomerular capillaries in each glomerulus every hour (Fig. 3b). Nearly 80% of these cells remained static (Fig. 3b), with an average dwell time of ~5 min (Fig. 3c). However, the minority

of MHCII$^+$ cells that crawled within glomerular capillaries were retained for more than twice as long (Fig. 3c). We confirmed these findings in glomeruli of intact, non-hydronephrotic kidneys of 3-week-old mice, in which ~15 adherent MHCII$^+$ cells underwent adhesion per glomerulus per hour, the majority of which were stationary (Fig. 3d, Supplementary Movie 6). These experiments demonstrate that MHCII-expressing leukocytes constitutively undergo periods of retention, and in some cases migration, in glomerular capillaries, even in the absence of inflammatory stimuli. Together with our observations of constitutive intraglomerular migration of CD4$^+$ T cells and altered retention of these cells in the presence of cognate antigen, these findings raise the possibility that circulating MHCII$^+$ leukocytes present glomerular antigens to intravascular effector CD4$^+$ T cells.

**T cells interact with intravascular antigen-presenting cells**. For intravascular MHCII-expressing leukocytes to present antigens to CD4$^+$ T cells within the glomerulus, these two populations must interact within glomerular capillaries. To determine if this occurs, CMTPX-labeled OT-II cells were imaged 30–90 min after transfer into MHCII-EGFP mice injected with either 8D1/pOVA or unconjugated 8D1. In both groups of mice, OT-II cells and MHCII$^+$ leukocytes consistently underwent interactions in the glomerular microvasculature (Fig. 4a, Supplementary Movie 7). On average, each OT-II cell interacted with one MHCII$^+$ cell during its time within the glomerulus (Fig. 4b), with the number of interactions being similar in mice receiving either 8D1 or 8D1/pOVA. The majority of interactions were no longer than 4 min. However, we noted more prolonged interactions (>10 min) in

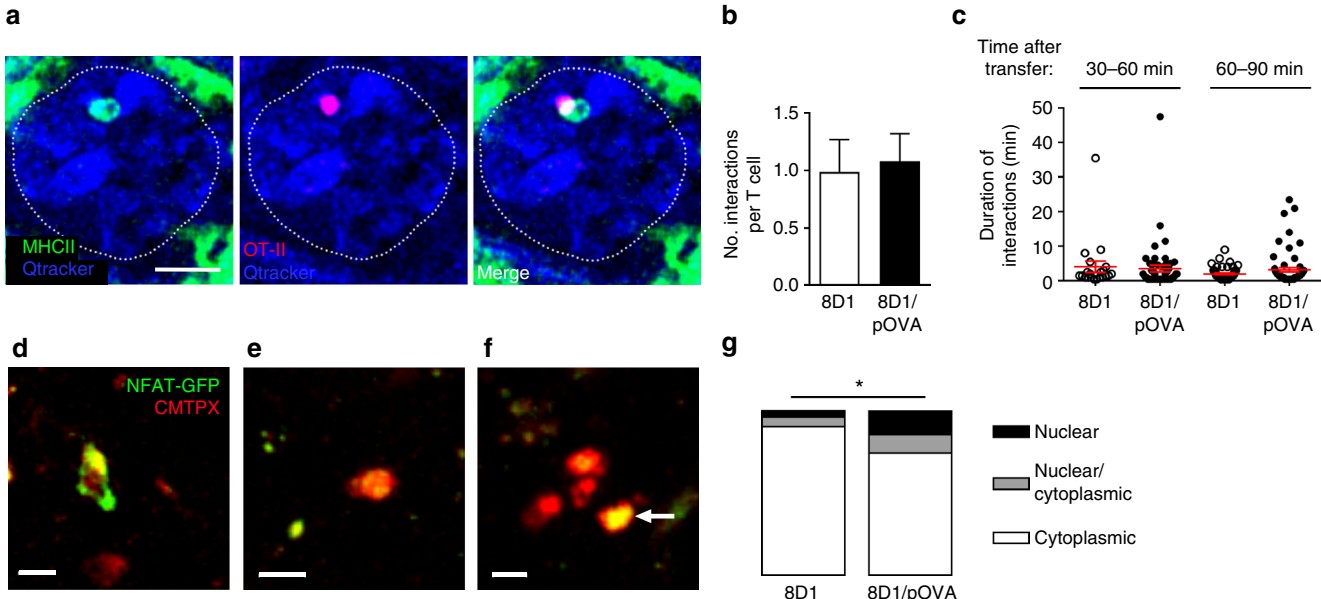

**Fig. 4** CD4$^+$ T cells interact with antigen-presenting cells in glomeruli and undergo activation. **a–c** OT-II cells were transferred into MHCII-EGFP mice with either unconjugated 8D1 or 8D1/pOVA and the interactions of OT-II cells and MHCII-expressing immune cells in the glomerular microvasculature were assessed by intravital multiphoton microscopy. **a** Multiphoton microscopy images of an MHCII-EGFP$^+$ leukocyte (green, left panel) interacting with a CMTPX-labeled OT-II cell (red, middle panel) in the glomerulus (vasculature detected by Qtracker® 655—blue). Right panel shows the merged image. Glomerular border is denoted by a thin dotted line. See also Supplementary Movie 7. Scale bar represents 10 μm. **b–c** Analysis of the intraglomerular interactions of OT-II cells and MHCII$^+$ leukocytes 30–90 min post-transfer. Interactions were defined as an adherent intraglomerular OT-II cell being immediately adjacent to an MHCII$^+$ cell for ≥30 s. Data are shown as mean ± s.e.m. (8D1, $n = 33$ cells from 10 mice; 8D1/pOVA, $n = 60$ cells from 11 mice). Data are shown for number (**b**) and duration (**c**) of OT-II cell/MHCII-EGFP$^+$ leukocyte interactions, assessed separately 30–60 min and 60–90 min after OT-II cell transfer, expressed per OT-II cell and shown as mean ± s.e.m. **d–g** Use of OT-II$_{NFAT-GFP}$ cells to demonstrate antigen-dependent T-cell activation in glomerular capillaries. **d–f** Images of OT-II$_{NFAT-GFP}$ cells in vivo in glomerular capillaries, illustrating OT-II cells with NFAT-GFP localized to the cytoplasm (**e**), nucleus and cytoplasm (**f**) or exclusively in the nucleus (arrow) (**g**). The cytoplasm is visible via CMTPX staining (red) while NFAT-GFP is visible in green. Scale bars denote 10 μm. See also Supplementary Movie 8. **f** Graphs showing percentages of T cells with NFAT-GFP in cytoplasmic, nuclear/cytoplasmic and nuclear locations, shown for mice treated with either 8D1 or 8D1/pOVA. Data show analysis of 51 cells from three experiments (8D1) and 89 cells from four experiments (8D1/pOVA). *$P < 0.05$ via Fisher's exact test, comparing the number of cells with cytoplasmic NFAT vs the combined total of the nuclear/cytoplasmic and nuclear categories in the two treatment groups

mice that received antigen (Fig. 4c), generally occurring 60–90 min after transfer. These results demonstrate that intravascular OT-II cells are able to interact with antigen-presenting cells within the glomerular capillaries.

**Antigen-dependent activation of OT-II cells in glomerulus.** To assess whether OT-II cells undergo activation within the glomerular capillaries, we made use of a system enabling visualization of translocation of the transcription factor nuclear factor of activated T cells-1 (NFAT1) to the nucleus[11,43]. In T cells, NFAT1 is a critical transcription factor driving changes in gene expression in response to T-cell activation, and rapidly translocates to the nucleus in response to elevation in cytosolic calcium following TCR engagement[44]. Via use of the fluorescent reporter NFAT1$_{(1–460)}$-GFP (NFAT-GFP), NFAT1 translocation to the nucleus, and therefore T-cell activation, can be tracked in vivo using multiphoton microscopy. We generated NFAT-GFP-transduced OT-II (OT-II$_{NFAT-GFP}$) cells and demonstrated in vitro that the NFAT-GFP reporter readily translocated to the nucleus following T-cell activation (Supplementary Fig. 3). We then transferred these cells into mice 30 min after treatment with either 8D1 or 8D1-pOVA and assessed subcellular localization of the reporter (Fig. 4d–f and Supplementary Fig. 3e–g). Following administration of 8D1-pOVA, NFAT-GFP had either partially or completely translocated to the nucleus in 26% of OT-II$_{NFAT-GFP}$ cells, whereas in mice that received 8D1, translocation was seen in

only 10% of cells ($P < 0.05$, Fisher's exact test) (Fig. 4g). Furthermore, typically T cells with cytoplasmic NFAT-GFP were active and highly migratory, while cells displaying nuclear NFAT-GFP remained static (Supplementary Movie 8), consistent with previous descriptions of T-cell arrest during antigen-dependent activation[31]. These experiments provide direct evidence of T-cell activation occurring within the glomerular capillaries.

To investigate whether OT-II cells in the kidney showed other changes typically associated with antigen recognition, 4 h after transfer kidneys were digested and the T cells analyzed by flow cytometry, assessing IFNγ production. In mice that received 8D1/pOVA, renal OT-II cells had significantly increased de novo production of IFNγ, compared with OT-II cells from mice that received 8D1 control mAb (Fig. 5a–c). This response was not seen in mice that received pOVA conjugated to a control antibody (IgG/pOVA) (Fig. 5b, c), demonstrating that pOVA administered in a non-targeted form was insufficient to induce this response.

We previously observed increased glomerular neutrophil retention in mice 4 and 24 h after receiving OT-II cells and 8D1/pOVA[17]. Here we confirmed that this response was also dependent on glomerular targeting of pOVA. Comparison of neutrophil responses 24 h after administration of OT-II cells plus either 8D1, 8D1/pOVA or non-targeted IgG/pOVA revealed that increased neutrophil dwell time was observed in mice that received 8D1/pOVA but not IgG/pOVA (Fig. 5d). As an additional readout of the neutrophil response, we used the oxidant-sensitive fluorochrome, DHE, to assess ROS production

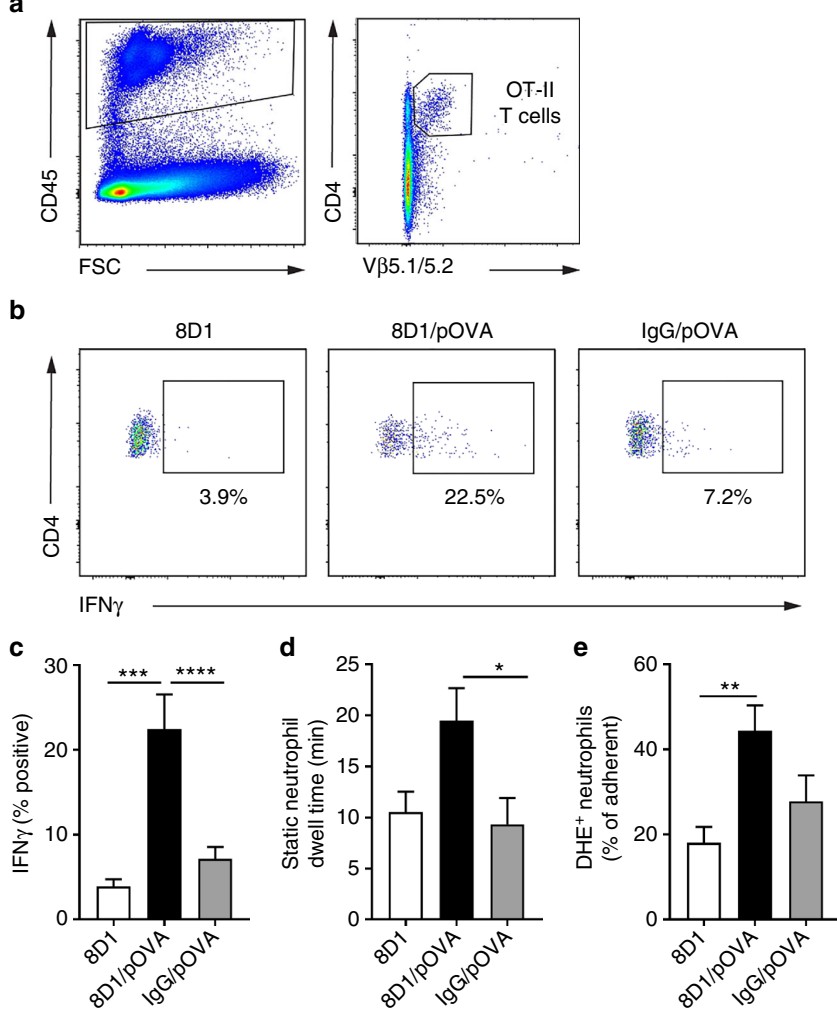

**Fig. 5** Glomerular targeting of pOVA promotes T-cell activation and glomerular inflammation. **a–c** Assessment of OT-II cell activation in the kidney in the presence of glomerulus-targeting antibody alone (8D1), glomerulus-targeted antigenic peptide (8D1/pOVA), or non-targeted antigenic peptide (IgG/pOVA). **a, b** Example flow cytometry data showing gating strategy to detect OT-II cells (left panel—forward scatter/CD45+; right panel—CD4+ Vβ5.1/5.2+) in kidney digests (**a**), followed by intracellular staining for IFNγ in all three experimental conditions (**b**). **c** Group data showing IFNγ expression (as % of OT-II cells) from mice receiving either 8D1, 8D1/pOVA or IgG/pOVA along with OT-II cells. Data are shown as mean ± s.e.m. of $n = 7$ (8D1) or 8 (8D1/pOVA and IgG/pOVA) mice per group. ***$P < 0.001$; ****$P < 0.0001$, via one-way ANOVA with Dunnett's multiple comparison test. **d, e** Neutrophil retention and activation in the glomerulus in the presence of 8D1, 8D1/pOVA or IgG/pOVA, as assessed by multiphoton microscopy. Data are shown for static dwell time (**d**) and % of adherent neutrophils that were DHE+ (**e**). Data are shown as mean ± s.e.m. of $n = 8$ (8D1 and IgG/pOVA) or 10 (8D1/pOVA) mice per group. *$P < 0.05$; **$P < 0.01$, via one-way ANOVA with Tukey's multiple comparison test (**d, e**)

by intraglomerular neutrophils[17,18]. These experiments revealed that induction of neutrophil ROS production also required glomerular targeting of antigen (Fig. 5e). Together these data indicate that glomerular localization of pOVA results in T-cell activation in the microvasculature of the glomerulus and induction of neutrophil-dependent glomerular inflammation.

**B cells are not required for T-cell-dependent inflammation.** We next sought to identify the MHCII-expressing immune cell in the circulation responsible for intravascular antigen presentation to CD4+ T cells in glomerular capillaries. Flow cytometric analysis of blood from MHCII-EGFP mice revealed that the majority (~95%) of MHCII-EGFP+ leukocytes were CD19+ B cells (Supplementary Fig. 4a). The remainder consisted primarily of a subset of CD115+ monocytes, while <1% of the MHCII+ population were CD11c+ dendritic cells. To determine whether B cells are retained in glomerular capillaries, we used anti-B220 to label

B cells in vivo (Supplementary Fig. 4b, Supplementary Movie 9). B cells adhered within glomerular capillaries at a rate of ~13 cells per glomerulus per hour (Supplementary Fig. 4c). The majority of adherent B cells remained stationary (Supplementary Fig. 4c), with a dwell time of ~8 min, while the minor population of crawling B cells had a shorter dwell time of only ~3 min (Supplementary Fig. 4d).

To determine whether B cells were required for induction of T-cell-mediated glomerular inflammation, we next assessed neutrophil retention in B-cell-deficient μ MT mice 4 h after transfer of OT-II cells and either 8D1/pOVA or unconjugated 8D1. μ MT recipient mice that received 8D1/pOVA had prolonged retention of neutrophils in glomeruli, compared with mice that received 8D1 (Supplementary Fig. 4e), similar to the response described previously in wild-type mice[17]. These findings indicate that while B cells comprise the majority of MHCII+ cells retained in the uninflamed glomerulus, they are not required for induction of CD4+ T-cell-mediated inflammation.

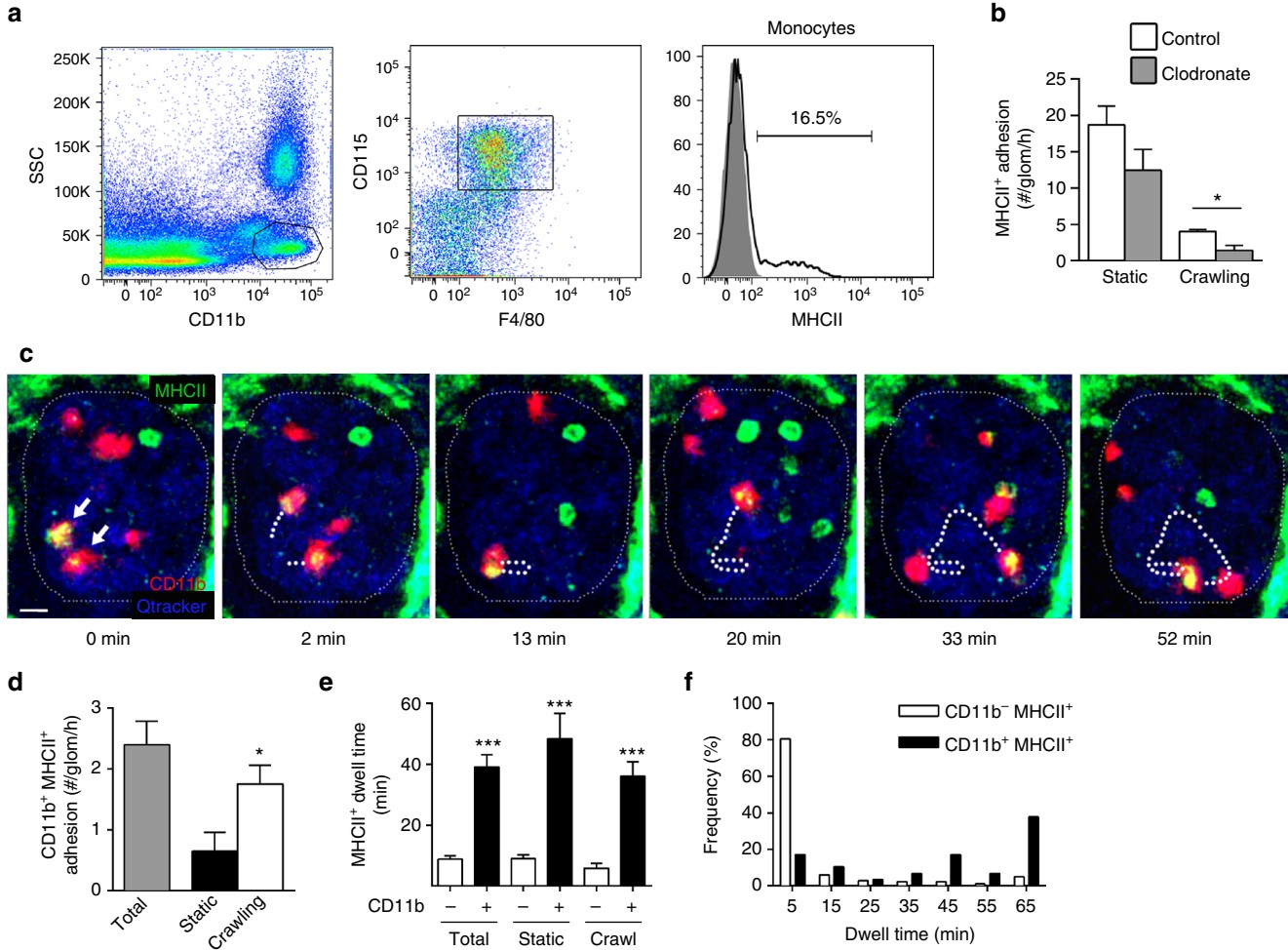

**Fig. 6** MHCII-expressing monocytes have prolonged retention in the glomerular microvasculature. **a** The proportion of monocytes expressing MHCII was determined by flow cytometric of circulating leukocytes from MHCII-EGFP mice. The SSC[lo] CD11b[hi] population was gated (left panel), then monocytes were identified as CD115[+] F4/80[+] (middle panel) and the proportion of MHCII-EGFP[+] monocytes gated on a histogram (right panel, black line) against a fluorescence-minus-one control (shaded). Results shown are representative of analyses from three mice. **b** Effect of clodronate liposomes on adhesion of MHCII-EGFP[+] leukocytes in glomerular capillaries assessed by intravital multiphoton microscopy analysis of MHCII-EGFP[+] leukocytes 18 h after treatment with either clodronate (gray bars) or control liposomes (white bars). Data show number of adherent cells defined as either static or crawling and represent mean ± s.e.m. from $n = 6$ mice per group. *$P < 0.05$ vs Control. **c**–**f** Intraglomerular migration of MHCII-EGFP[+] CD11b[+] cells as assessed by intravital multiphoton microscopy. **c** Multiphoton microscopy image sequence showing the migration paths of two MHCII-EGFP[+] CD11b[+] leukocytes (yellow cells, white arrows). Dotted lines show the cell migration paths. Time elapsed is indicated below panels. See also Supplementary Movie 10. **d** Number of adherent MHCII-EGFP[+] CD11b[+] cells, shown for total cells, and separately for static and crawling cells. *$P < 0.05$ vs 'Static'. **e** Dwell time for CD11b[−] MHCII[+] and CD11b[+] MHCII[+] cells, shown for total cells and separately for static and crawling cells. ***$P < 0.001$ vs corresponding CD11b[−] MHCII-EGFP[+] populations. **f** Frequency distribution of the dwell times of CD11b[−] MHCII-EGFP[+] vs CD11b[+] MHCII-EGFP[+] cells. Data in **d**–**f** were derived from $n = 4$ mice (CD11b[+] $n = 29$ cells; CD11b[−], $n = 181$ cells) and are shown as mean ± s.e.m. (**d**, **e**). **b**, **d**, **e** Mann−Whitney or unpaired Student's t-tests

**MHCII[+] monocytes have prolonged intraglomerular retention.** We next examined the contribution of circulating monocytes to the MHCII[+] leukocyte population undergoing retention in glomerular capillaries. Flow cytometric analysis of blood from MHCII-EGFP mice revealed that 10–20% of circulating monocytes expressed MHCII (Fig. 6a) similar to previous reports[42,45,46], with these cells being found in both CX3CR1[lo-int] (classical) and CX3CR1[hi] (non-classical) populations (Supplementary Fig. 5). To determine whether monocytes contributed to the MHCII[+] leukocyte population patrolling the glomerulus, we treated MHCII-EGFP mice with clodronate liposomes and analyzed migration of EGFP[+] cells by multiphoton microscopy. Clodronate liposome treatment has recently been shown to almost eliminate patrolling monocytes from the glomerulus[18]. In MHCII-EGFP mice, clodronate liposomes did not significantly affect the number of stationary MHCII[+] cells in glomerular

capillaries (Fig. 6b), consistent with the continued retention of B cells. In contrast, the number of MHCII[+] leukocytes undergoing crawling was significantly decreased after clodronate liposome treatment, showing that monocytes constitute the majority of the crawling MHCII[+] leukocyte population within glomerular capillaries.

In the absence of inflammation, monocytes extensively patrol the glomerular microvasculature, being retained in the glomerular capillaries for an average of ~20 min[17,18]. To determine whether MHCII[+] monocytes were retained for a similar extended time, we utilized anti-CD11b to detect monocytes in MHCII-EGFP mice[3]. As B cells may also express CD11b, we first performed flow cytometric analyses of whole blood, demonstrating that the majority (>80%) of CD11b[hi] MHCII[+] leukocytes were monocytes, while <10% were B cells (Supplementary Fig. 6). Using intravital multiphoton microscopy of the glomerular capillaries,

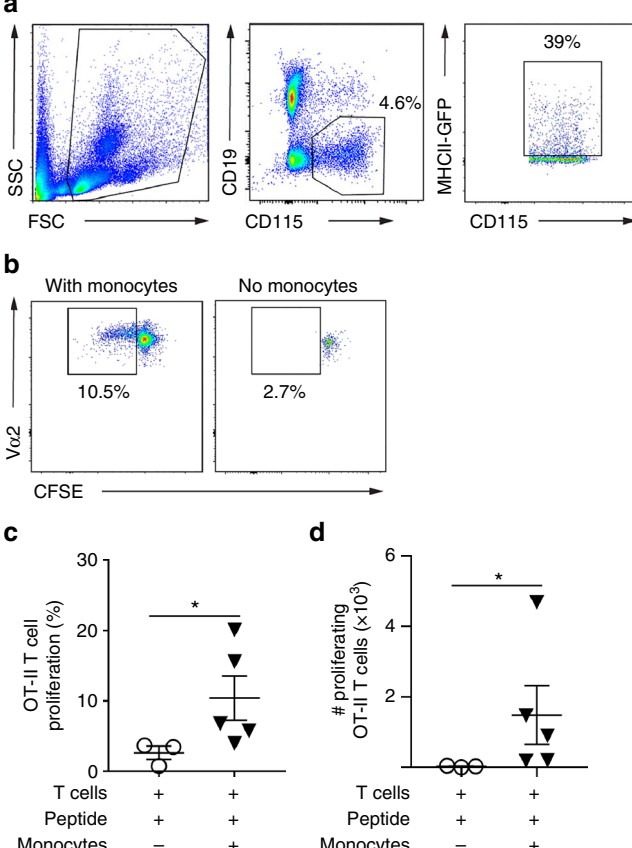

**Fig. 7** MHCII+ monocytes are capable of inducing CD4+ T-cell proliferation. MHCII+ monocytes were sorted from the blood of MHCII-GFP mice and used as antigen-presenting cells in in vitro CFSE dilution T-cell proliferation assays with OT-II cells. **a** Flow cytometry plots showing gating strategy used to isolate MHCII+ monocytes, initially gating on forward and side scatter profiles (left panel), then identifying monocytes (CD19− CD115+—central panel), then selecting GFP+ cells within the monocyte population (right hand panel). **b–d** OT-II cell proliferation as assessed via CFSE dilution. **b** Representative FACS plots of proliferating OVA-specific Vα2+ OT-II CD4+ T cells (gated as shown in Supplementary Fig. 7) shown for T cells cultured with peptide-pulsed MHCII+ monocytes vs T cells with peptide alone. CFSE dilution (proliferation) is observed in the presence of MHCII+ monocytes but not in the absence of monocytes. Inset numbers represent the average proportion of cell populations over five individual experiments. **c**, **d** Group data from individual experiments performed with T cells and OVA peptide alone, or with T cells, OVA peptide and MHCII+ monocytes, showing OT-II cell proliferation as % (**c**) and number (**d**) of OT-II cells. *$P < 0.05$ via Mann–Whitney tests

we observed retention of three populations: (1) MHCII+ CD11b− cells, the bulk of which were likely to be B cells; (2) MHCII− CD11b+ cells (neutrophils or monocytes); and (3) MHCII+ CD11b+ cells (Fig. 6c and Supplementary Movie 10). Adhesion of these CD11b+ MHCII+ leukocytes occurred at a rate of ~2.5 cells per glomerulus per hour, with the majority undergoing crawling (Fig. 6d). Interestingly, the mean dwell time of CD11b+ MHCII+ cells in glomeruli was ~40 min, compared to <10 min for CD11b− MHCII+ cells, with this applying to both crawling and stationary cells (Fig. 6e). Furthermore, cells with dwell times >35 min were almost exclusively CD11b+ MHCII+ (Fig. 6f). As B cells typically have a short (~6 min) dwell time in glomeruli (Supplementary Fig. 4d), these findings support the interpretation that the CD11b+ MHCII+ leukocytes with prolonged dwell times are not B cells.

MHCII+ monocytes are not recognized as professional antigen-presenting cells. Nevertheless, for them to be important in T-cell-mediated glomerular inflammation, they must be able to induce antigen-dependent responses in T cells. Therefore, we sorted MHCII+ monocytes from the blood of MHCII-GFP mice (Fig. 7a) and assessed their capacity to induce OT-II cell proliferation in vitro[39,40]. In the presence of OVA-peptide, MHCII+ monocytes were able to induce OT-II T-cell proliferation (Fig. 7b–d). T cells did not proliferate in the absence of monocytes and the level of T-cell proliferation was proportional to the number of monocytes in the assay, supporting the specificity of this response. These observations clearly demonstrate that the MHCII-expressing monocyte subset can present antigen to CD4+ T cells. Finally, using electron microscopy, we observed that mononuclear immune cells patrolling glomerular capillaries extended projections in close proximity to fenestrae of glomerular endothelial cells (Supplementary Fig. 8), raising the possibility that these leukocytes were probing the location of the antigen in these experiments. Together these findings indicate that a population of MHCII+ monocytes undergoes constitutive patrolling within the glomerular microvasculature, characterized by substantially longer intraglomerular dwell times than any other population previously identified, and that MHCII+ monocytes have the capacity to induce CD4+ T-cell proliferation.

**Monocytes are required for T-cell-induced inflammation.** We next assessed whether the presence of MHCII+ patrolling monocytes within glomeruli was associated with a role for monocytes in induction of antigen-dependent glomerular inflammation, using clodronate liposomes to deplete monocytes in the OT-II cell-8D1/pOVA model. Firstly we established that OT-II cell trafficking in the glomerulus was unaltered in clodronate-treated mice (Supplementary Fig. 9). Subsequently we used neutrophil retention and ROS production in glomerular capillaries as readouts of T-cell-initiated glomerular inflammation following clodronate treatment. In mice depleted of monocytes, both neutrophil dwell time and ROS production 24 h after transfer of OT-II cells and 8D1/pOVA were reduced relative to mice treated with control liposomes (Fig. 8a–c and Supplementary Movie 11), with the reduction in ROS-producing neutrophils due to reduced production by crawling cells (Fig. 8d). Together these findings support the hypothesis that induction of glomerular inflammation by intravascular OT-II cells in response to planted antigen requires monocytes.

## Discussion
CD4+ T cells specific for nephritogenic autoantigens are present in the circulation during autoimmune glomerulonephritis[26–28], and experimental evidence indicates that CD4+ T-cell-dependent glomerular inflammation can be initiated by T cells responding to antigen located within the vasculature[15,16]. However, the cell type that presents antigen to effector CD4+ cells within glomeruli is not known. The present study addresses this by demonstrating that the cellular participants required for a CD4+ T-cell-mediated response—CD4+ T cells and an MHCII-expressing cell—both undergo constitutive periods of intravascular retention and crawling in the glomerular microvasculature. In addition, here we observed that each T cell retained in the glomerulus interacts with on average one MHCII-expressing immune cell. In the presence of antigen, T cells in the glomerular capillaries displayed evidence of antigen recognition including nuclear translocation of NFAT1, prolonged dwell time and reduced migration velocity. The MHCII-expressing cells retained in the glomerulus for the longest duration expressed the monocyte marker CD11b and monocyte

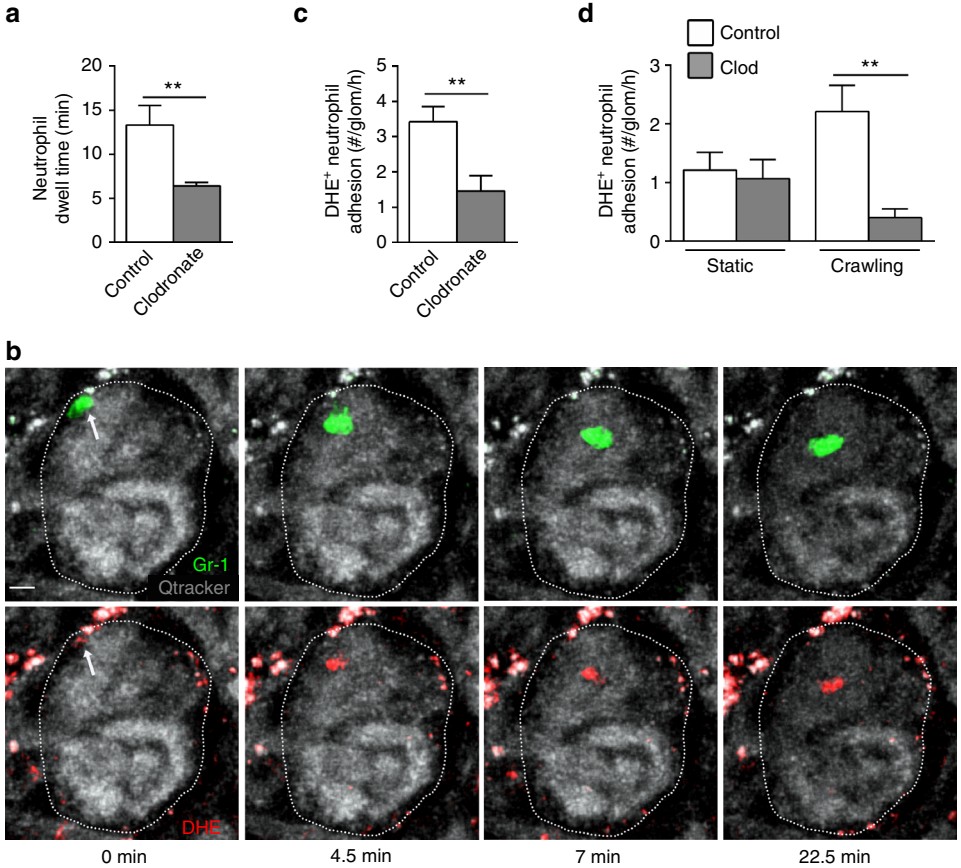

**Fig. 8** Monocytes are required for T-cell-induced neutrophil activation in response to planted antigen. The effect of clodronate liposome-mediated monocyte depletion on glomerular neutrophil retention and activation was assessed 24 h after transfer of OT-II cells and 8D1/pOVA via intravital multiphoton microscopy. Mice were treated with either control liposomes (white bars) or clodronate liposomes (gray bars) 18 h prior to cell transfer. **a** Neutrophil dwell time in control and clodronate liposome-treated mice. **b–d** Production of reactive oxygen species (ROS) by neutrophils, as assessed in intravital microscopy experiments via the ROS-sensitive fluorochrome, dihydroethidium (DHE). **b** Multiphoton image sequence showing a neutrophil (arrow) (identified by Gr-1 staining, green, upper panels) positively stained for DHE (red, lower panels). Time elapsed is shown beneath each panel. See also Supplementary Movie 11. **c** Number of DHE⁺ neutrophils adherent in the glomerular capillaries shown for the total population (**c**) and separately for static and crawling neutrophils (**d**). Data are shown as mean ± s.e.m. from seven mice per group. **P < 0.01 vs Control. **a**, **d** Mann−Whitney tests; **c** unpaired Student's unpaired t-test

depletion attenuated inflammation induced by T-cell antigen recognition. Together these findings demonstrate that a CD4⁺ T-cell-driven immune response in the glomerulus can be mediated by intravascular antigen presentation by an MHCII-expressing subset of circulating monocytes.

We hypothesized that a key step in the development of T-cell-mediated glomerulonephritis occurs when T cells recognize disease-causing antigens within the glomerular microvasculature. Here we provide direct evidence of T-cell activation in the glomerulus during the initial response to locally planted antigen. Using the NFAT-GFP reporter to reveal T-cell activation in vivo at a single cell level, we visualized increased nuclear translocation of this transcription factor in T cells in glomerular capillaries, in the presence of cognate antigen targeted to the glomerulus. These findings clearly demonstrate that antigen-dependent T-cell activation occurs intravascularly within the glomerular capillaries. Glomerular targeting of pOVA was necessary for both T-cell activation and downstream intraglomerular neutrophil activation, as these responses were not seen when pOVA was delivered systemically in a non-targeted fashion. Together these data provide compelling evidence that T cells respond to antigen within glomerular capillaries and that intraglomerular antigen recognition leads to proinflammatory changes in the glomerular microvasculature. In some studies, immunogenic peptides capable of

binding directly to MHCII without intracellular uptake or processing have been used to examine the immediate response to antigen presentation[31]. As our studies use the peptide recognized by OT-II cells, it is conceivable that a similar response could be at work here. However, as the peptide is covalently conjugated to a much larger immunoglobulin molecule, we anticipate that intracellular processing is the more likely route by which the OVA peptide is loaded onto MHCII.

In this planted antigen model of T-cell-dependent glomerular inflammation, we reasoned that the cell responsible for antigen presentation must either be intravascular or have access to the glomerular capillary lumen. The renal interstitium plays host to an abundant mononuclear phagocyte population[39,41,47,48], and these cells may access the lumen of the interstitial cortical renal microvasculature via extension of cellular processes[49,50]. However, results of the present experiments indicate that this is not the case in the glomerulus. Furthermore, no intrinsic glomerular cells, including glomerular endothelial cells, expressed detectable MHCII under resting conditions. Therefore, we next examined circulating immune cells with the potential to present antigen to CD4⁺ T cells. Circulating B cells expressed MHCII. However, OT-II cells could induce glomerular inflammation in the absence of B cells. Therefore we examined the possibility that a monocyte subpopulation mediated this response.

In mice, monocytes exist in two major subpopulations: classical/inflammatory (CCR2[hi] CX₃CR1[lo] Ly6C[+]) and non-classical/patrolling (CCR2[−] CX₃CR1[hi] Ly6C[−]). Mounting evidence indicates that the patrolling subset has a specialized role in mediating responses to injury or infection from within the vasculature, including in the glomerulus[2,3,18,51], and these cells also undergo prolonged retention and migration in the glomerular microvasculature[17,18]. We observed that a subset of the MHCII-expressing cells that were positive for the monocyte marker CD11b were retained in the glomerulus, often for over 40 min. This prolonged patrolling of the glomerular microvasculature provides opportunity to encounter and internalize antigens while also increasing the probability of this cell encountering intraglomerular T cells. These properties are consistent with an immunosurveillance function for these cells. We also demonstrate that MHCII[+] monocytes can induce antigen-specific T-cell proliferation in vitro, providing additional support for the contention that these cells are responsible for antigen-specific T-cell activation in glomerular capillaries. With future technical advances, it may be feasible to isolate or image patrolling MHCII[+] monocytes in glomeruli to demonstrate antigen uptake from within the glomerular microvasculature.

To examine a role for monocytes in T-cell-induced glomerular inflammation, we used clodronate to remove intravascular monocytes. Clodronate did not affect retention of the MHCII-expressing cells that underwent brief, static retention in glomeruli, consistent with our observations that these were B cells. In contrast clodronate markedly reduced the number of migratory MHCII-expressing cells present in glomeruli, providing further evidence that these cells were monocytes. This depletion strategy resulted in a reduction in the downstream inflammatory readouts of neutrophil retention and ROS-generating activity, consistent with the hypothesis that MHCII-expressing monocytes are responsible for intravascular antigen presentation in this model. We recently reported that monocytes promote innate responses of intravascular neutrophils in acute glomerulonephritis[18]. The present findings add to this by showing that monocytes can also perform the function of intravascular antigen presentation to T cells in T-cell-mediated glomerular injury. These findings are of particular relevance to the initial stages of disease. However in established autoimmune glomerulonephritis, further mechanisms of antigen recognition, including antigen presentation by glomerular endothelial cells with upregulated MHCII expression could contribute to activation of CD4[+] T cells[52].

Alterations in migration are characteristic of T cells undergoing antigen recognition. In naïve cells, altered migration occurs over several hours[53,54], but effector T cells respond to antigen presentation rapidly via migratory arrest and induction of cytokine production[31,32]. The T-cell responses in the present study are consistent with previous descriptions of effector T cells. Identification of cells undergoing antigen presentation on the basis of nuclear localization of NFAT-GFP revealed that these cells were uniformly static, while cells with non-translocated NFAT-GFP, and therefore not detecting antigen, continued to migrate. This T-cell response occurred within the first hour after T-cell transfer. Analysis of cytokine production provided further evidence of rapid antigen-specific activation by intrarenal T cells in that many intrarenal effector CD4[+] T cells were producing IFNγ 4 h after transfer. Finally, changes in neutrophil retention and activation resulting from this rapid T-cell response were apparent within 4 h. These findings highlight the speed with which effector T cells can respond to antigen in glomeruli and promote downstream inflammation. It should be noted that in these experiments, we used effector T cells generated via a standard ex vivo differentiation protocol. It is conceivable that responses of cells generated in this manner may differ from those

of cells differentiated in vivo, a possibility that could be explored in future studies.

The actions of neutrophils observed in glomerular capillaries under resting conditions are similar to those reported for neutrophils in uninflamed pulmonary capillaries[55–57]. In the lung, this intravascular patrolling may be a form of immune surveillance facilitating rapid responses to local infection. It is possible that this is also an important function of neutrophils in glomeruli. However, microbial infection is less explicitly relevant in the glomerular microcirculation than in the lung. In the glomerulus neutrophil patrolling is a double-edged sword as it also underpins the induction of injurious responses after immune complex deposition or in response to other inflammatory stimuli[17].

In conclusion, these studies have identified a novel mechanism of intravascular antigen recognition in the glomerular microvasculature, in which circulating CD4[+] T cells routinely interact with patrolling antigen-presenting cells. Under these circumstances, deposition of antigen within the glomerular vasculature can result in rapid induction of inflammation in the glomerular capillaries. This is the first study to show the coordinated series of events underlying initiation of intravascular CD4[+] T-cell responses in the glomerulus.

## Methods

**Mice.** C57BL/6 wild-type mice were obtained from Monash Animal Research Platform at Monash University and housed in specific pathogen-free conditions. MHC Class II-EGFP knock-in (MHCII-EGFP) mice on a C57BL/6 background, generously provided by B. Fazekas de St Groth (University of Sydney) were bred in-house[37]. μ chain-deficient (μMT) mice, CD11c-YFP mice, Cx3cr1[gfp/+] mice, and OT-II mice, all on a C57BL/6 background, were bred in-house. SMARTA-GFP mice were generously provided by S. Mueller (University of Melbourne). Male mice between 3 and 22 weeks of age were used in all experiments. All experimental procedures were approved by the Monash Medical Centre Animal Ethics Committee 'B'. Sample size calculation was not performed a priori because the effect sizes of our observations and interventions could not be determined before experiments. This study was not randomized and was not blinded. All experiments were included in the analyses.

**Antibodies and reagents.** For production of ovalbumin peptide (pOVA)-conjugated 8D1, we used 8D1 mAb[15,33] (grown from hybridoma), chemical linker N-succinimidyl-6-maleimido-caproate (EMCS, Sigma Aldrich), and a custom OVA₃₂₃₋₃₃₉ peptide containing an amino terminal cysteine residue and C-terminal biotin tag (Mimotopes, Notting Hill, VIC, Australia). As control for these experiments, isotype control mouse IgG1 (clone MOPC-21, Bio X Cell)[58] was used. For activation and polarization of OT-II T cells, cells were cultured in RPMI 1640 supplemented with 10% heat-inactivated fetal calf serum (both from Life Technologies), 2-mercaptoethanol, recombinant mouse (rm)IL-12 (eBioscience), rat anti-IL-4 (clone 11B11, ATCC) and rmIL-2 (R&D Systems). Mitomycin C was from Intas Pharmaceuticals (Hyderabad, India), Histopaque 1083, Optiprep and Brefeldin A were acquired from Sigma, and collagenase D and DNase I were from Roche Diagnostics (Dee Why, NSW, Australia). For mouse anesthesia, ketamine hydrochloride (Troy Lab) and xylazine (Pfizer) were used.

For in vivo imaging experiments, the following mAbs were used: anti-CD4 PE (clone GK1.5, 2 μg)[59], anti-B220 PE (RA3-6B2, 2 μg)[60] (BD Pharmingen), and anti-CD11b NC650 (clone M1/70, 10 μg)[3], anti-Gr-1 PE and anti-Gr-1 Alexa488 (both RB6-8C5, 2 μg)[17] (eBioscience). Dihydroethidium (DHE), carboxyfluorescein diacetate succinimidyl ester (CFSE), Cell Tracker Red (CMTPX), Qtracker® 655 and rhodamine dextran were purchased from ThermoFisher Scientific (Scoresby, VIC, Australia). Clodronate and control liposomes were purchased from ClodronateLiposomes.com (Amsterdam, The Netherlands). In some experiments, 8D1 conjugated to Pacific Blue (ThermoFisher Scientific) in-house was used as a vascular label.

The following mAbs were used for flow cytometry (see Supplementary Table 1 for information on antibodies used in this study): anti-CD115 APC (clone AFS98) and anti-CD45 APC (30-F11) (eBioscience); and anti-CD19 APC-Cy7 (1D3), anti-CD11c APC (HL3), anti-CD11b APC-Cy7 (M1/70), anti-Ly6C-APC (AL-21), anti-CD4 APC-Cy7 (GK1.5), anti-IFNγ PE (XMG1.2), anti-TCR Vβ5.1/5.2 FITC (MR9-4) (BD Pharmingen), anti-MHC Class II-PE (M5/114.15.2), anti-TCR Vα2 AF647 (B20.1) (grown from hybridoma and conjugated in-house)[39,40,61,62]. Gating strategies used to define the relevant populations are shown in each figure.

**In vitro activation of OT-II T cells.** OT-II T cells were differentiated into Th1 effector cells as described previously[15,17]. Briefly, lymph nodes were harvested from OT-II mice, and cells were isolated and cultured at 1×10⁶ cells mL⁻¹. Splenocytes

from C57BL/6 mice were treated with mitomycin C (50 µg mL$^{-1}$, 20 min at 37 °C), then washed thoroughly and cultured with naïve OT-II cells at a ratio of 10:1 in medium containing 1 µM OVA$_{323-339}$ peptide. Cells were maintained in media containing rmIL-12 (2 ng mL$^{-1}$) and anti-IL-4 mAb (11B11, 10 µg mL$^{-1}$). After 48 h, rmIL-2 (5 µg mL$^{-1}$) was added and maintained thereafter. Cells were split once and harvested on day 7. Cell preparations were centrifuged over Histopaque 1083 or Optiprep (1.09 g mL$^{-1}$) for 30 min at 400 g at room temperature to remove dead cells. 1×10$^7$ OT-II T cells were transferred into recipient mice via jugular or tail vein unless otherwise stated. For analysis of migration, OT-II T cells were labeled with 5 µM CFSE or 1 µM CMTPX prior to transfer. In some experiments, CD4$^+$ T cells from GFP-expressing SMARTA TCR transgenic mice (specific for the LCMV GP-derived P13 peptide) underwent the same activation protocol, using P13 peptide (1 µM) as the stimulating antigen[63]. In these experiments, SMARTA cells showed comparable levels of activation as OT-II cells (Supplementary Fig. 10).

**Antibody peptide conjugation.** The 8D1 mAb (of IgG1 subclass) was conjugated to OVA$_{323-339}$ using a previously published technique[15]. In brief, 8D1 mAb was mixed with tenfold molar excess of EMCS for 2 h at room temperature. Unreacted EMCS was removed by buffer exchange chromatography, and the activated 8D1 was mixed with a tenfold molar excess of modified OVA$_{323-339}$. After incubation for 3 h at room temperature, the reaction was halted by adding 2 mM cysteine, and the modified 8D1/pOVA antibody was dialyzed in PBS to remove excess OVA$_{323-339}$. For some control experiments, pOVA was conjugated to an IgG1 control antibody of irrelevant specificity (MOPC-21) using the identical protocol.

**Induction of T-cell-dependent glomerular inflammation.** In order to specifically localize OVA$_{323-339}$ to the glomerular vasculature, the 8D1/OVA$_{323-339}$ conjugate was used[15,17], taking advantage of the selective binding of 8D1 to the NC1 domain of α3(IV) collagen in the glomerular basement membrane[33]. To induce glomerular inflammation, 150 µg of 8D1/pOVA was transferred intravenously into recipient mice together with OT-II T cells. Unconjugated 8D1 mAb and pOVA conjugated to MOPC-21 ("IgG/pOVA") served as controls.

**Renal intravital multiphoton microscopy.** To prepare the kidney for glomerular intravital microscopy, 4–5-week-old mice underwent unilateral ureteric ligation[17,18]. Mice were housed for 12 weeks to allow the kidney to undergo hydronephrosis. Intravital imaging experiments performed on intact kidneys were carried out in 3–4-week-old mice.

Multiphoton microscopy was used for intravital imaging studies[17,18]. Mice were anesthetized by an initial intraperitoneal injection of 150 mg kg$^{-1}$ ketamine hydrochloride and 10 mg kg$^{-1}$ xylazine. The left jugular vein was catheterized to allow delivery of fluorescent probes and to maintain anesthesia as required. A heating pad was used to maintain the temperature of the mice at 37 °C. The hydronephrotic kidney was exteriorized through a lateral incision and drained of urine using a 30G needle. The kidney was extended over a heated platform and secured with 4-0 silk tied to the kidney capsule. The kidney was superfused with saline (0.9% sodium chloride (wt per vol)) and covered with a coverslip held in place with vacuum grease. For imaging experiments using intact kidneys, mice were anesthetized and catheterized as above. The left kidney was exteriorized through a dorsal incision and the kidney was immobilized in a heated well incorporated into a custom-built stage. The exposed kidney was bathed in normal saline and coverslipped.

Glomeruli were observed with a Leica SP5 multiphoton microscope (Leica Microsystems), equipped with 20× 1.0 NA WI objective lens and a MaiTai pulsed infrared laser (SpectraPhysics). Experiments in C57BL/6 and µMT mice were performed at 810 nm excitation, while experiments in MHCII-EGFP mice were performed at 900 nm excitation. In most experiments, images were taken every 30 s by collecting z-stacks of approximately 150 µm depth, with 6 µm step size. For 2 h imaging experiments, z-stacks were collected every 60 s. Emitted fluorescence was detected by non-descanned detectors with 432–482 nm, 500–550 nm, 575–605 nm, and 625–675 nm emission filters. Pre-defined settings for laser power and detector gain were used for all experiments. For visualization of the vasculature, either Qtracker® 655, rhodamine dextran or Pacific Blue-conjugated 8D1 was used. To label endogenous leukocytes, the following mAbs were used as appropriate (2 µg i.v. each unless otherwise stated): anti-CD4 PE (CD4$^+$ T cells); anti-B220-PE (B cells); anti-Gr-1 PE or Alexa488 (neutrophils); anti-CD11b NC650 (monocytes – 5 µL). OT-II cells were labeled with either CFSE or CMTPX according to the manufacturer's instructions. In experiments examining neutrophil ROS production, mice received 2 mg kg$^{-1}$ of pre-warmed DHE intravenously 20 min prior to imaging and were examined using 810 nm excitation[17].

**Image analysis.** Images were analyzed using Imaris software (Bitplane)[17]. In brief, leukocytes arrested in glomerular capillaries for at least 30 s (two consecutive frames) were defined as adherent and subsequently categorized as crawling or static. Dwell time was defined as the duration of leukocyte adhesion in the glomerulus. To measure the velocity of crawling cells, images were tracked in three dimensions over time. For neutrophil ROS production, neutrophil DHE positivity was determined after adjusting images using pre-determined thresholds to remove background staining.

**Renal leukocyte isolation and analysis.** For flow cytometry analysis of OT-II cell cytokine production, OT-II cells were isolated from kidneys using a modification of a previously published method[64]. Kidneys were harvested 4 h after transfer of OT-II T cells, infused with RPMI 1640 containing collagenase (1 mg mL$^{-1}$), DNase I (100 µg mL$^{-1}$) and Brefeldin A (10 µg mL$^{-1}$) and incubated at 37 °C for 25 min. Kidneys were gently dissociated and incubated at 37 °C for a further 25 min then washed and resuspended in RPMI with 5 µg mL$^{-1}$ Brefeldin A. The cell suspension was left for 10 min to settle tubular debris. The supernatant was filtered through a 70 µm cell strainer and erythrocytes were lysed. Cells were treated with Fc block, then stained for CD45, CD4, and Vβ 5.1/5.2 TCR. For intracellular cytokine staining, cells were fixed and permeabilized using the BD Cytofix/Cytoperm™ kit, as per the manufacturer's instructions, and then stained for IFN-γ.

**MHCII$^+$ monocyte isolation and ex vivo T-cell proliferation.** MHCII$^+$ monocytes were sorted from blood of MHCII-GFP donors. Mice (7–11 per experiment) were bled into heparinized syringes and blood lysed using NH$_4$Cl. Leukocytes were resuspended to 5×10$^6$ cells in 50µL then pre-incubated with anti-CD16/CD32 Ab to block non-specific staining. Cells were stained with CD115-APC (clone AFS98) to identify monocytes and anti-CD19-APC-Cy7 (clone 1D3) to enable exclusion of B cells. CD115$^+$ monocytes that also expressed MHCII-GFP were sorted by flow cytometry (FACS ARIA Fusion). Post-sort analysis revealed >98% monocytes of which >85% were MHCII$^+$.

OT-II cell proliferation was assessed using a modification of a previously published technique[39,40]. In brief, CD4$^+$ T cells were positively enriched from LNs of OT-II mice using anti-CD4 microbeads (Miltenyl-Biotec) (purity >98%) then labeled with CFSE (5 µM). Cells were subsequently washed twice and resuspended at 1×10$^6$ mL$^{-1}$. Sorted monocytes (ranging from 1.6 to 8.8 ×10$^4$ per well) were pulsed with 1 µM OVA$_{323-339}$ and co-cultured with 10$^5$ CFSE-labeled OT-II cells. Negative controls included T cells pulsed with peptide alone (no antigen-presenting cells) and unpulsed splenocytes (no peptide). As a positive control, splenocytes pulsed with antigen as above were used as antigen-presenting cells. Cells were harvested after 65 h and stained with anti-Vα2 APC (clone B20.1) and anti-CD4 APC-Cy7 (clone GK1.5). Dead cells were excluded by propidium iodide staining and viable cells were enumerated using BD Calibrate beads (BD Biosciences). Samples were acquired on an FACS CantoII flow cytometer and analyzed with FlowJo software (Tree Star), using the level of CFSE staining as an index of proliferation.

**NFAT-GFP vector transduction.** T-cell activation was investigated via assessment of nuclear mobilization of NFAT-GFP, using T cells transduced with retrovirus generated from pMSCVneo–ΔNFAT-GFP (aa 1–460 of mouse NFAT1) ("NFAT-GFP") vector (obtained from Dr. Vigo Heissmeyer), as described previously[11,43]. Briefly, pMSCVneo–ΔNFAT-GFP was transfected into Platinum E packaging cells (Cell Biolabs, Inc.) using a calcium phosphate precipitation method[65,66]. After 48–72 h, retroviral supernatants were collected and filtered through 0.45 µm filters. The virus was concentrated by precipitation using PEG 6000 and sodium chloride, resuspended in RPMI and titrated[66].

OT-II cells were transduced with 1.5–3.2×10$^7$ transducing units of virus on days 1 and 2 of the activation protocol, in six-well plates in the presence of polybrene (4 µg mL$^{-1}$), using spinfection at 32 °C, 800 g for 1 h. On day 7, transduced NFAT-GFP OT-II cells (OT-II$_{NFAT-GFP}$) were isolated via flow cytometric sorting on the basis of GFP expression. Typical transduction efficiencies were between 10 and 20%. Preliminary analyses demonstrated that following in vitro activation, the level of IFNγ expression in OT-II$_{NFAT-GFP}$ cells was comparable to that in activated non-transduced OT-II cells (Supplementary Fig. 3). Sorted OT-II$_{NFAT-GFP}$ cells were labeled with CMTPX and used in in vivo imaging experiments assessing the subcellular localization of GFP[11,43].

**Statistical analysis.** All data presented are shown as mean ± s.e.m. Experimental groups were compared using Student's unpaired t tests, or if variances were unequal, unpaired t-tests with Welch's correction or non-parametric Mann−Whitney tests (all one-tailed). In experiments involving more than two groups, one-way ANOVA or Kruskal−Wallis non-parametric analysis was performed. In experiments involving categorical analysis of T-cell phenotype, Fisher's exact test was used. The numbers of mice used in groups were based on the expected degree of variability observed in standard kidney imaging experiments, in parameters such as number of adherent cells and dwell time. Typical group sizes ranged from 6 to 8 individual mice examined on multiple different days, assigned randomly to control or treatment groups. Significance was set at $P < 0.05$.

**Data availability.** The data that support the findings of this study are available from the corresponding author on request.

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

## Acknowledgements

The authors would like to acknowledge Cecilia Lo for technical assistance, Prof. Barbara Fazekas de St Groth (University of Sydney) for provision of MHCII-EGFP mice, A/Prof. Scott Mueller (University of Melbourne) for provision of SMARTA mice, Dr. Vigo Heissmeyer (LMU, Munich) for provision of the NFAT-GFP vector, Michael Thomson (FlowCore, Monash University) for assistance with cell sorting and Dr. Sarah Creed, Dr. Georg Ramm and Adam Costin (Monash Micro Imaging) for assistance with live cell imaging and electron microscopy. This study was supported by funding from the National Health and Medical Research Council (NHMRC), Australia (Grant IDs 1045165 and 1064112, to M.J.H. and A.R.K.). M.J.H. is an NHMRC Senior Research Fellow (Grant ID 1042775).

## Author contributions

C.L.V.W., M.U.N., S.L.S. and S.L. designed, performed and analyzed the experiments. P.H., M.F., A.L. and Z.H.T. performed and analyzed the experiments. C.L. and S.K.N. provided analytical tools. A.R.K. and M.J.H. designed the experiments and wrote the paper.

## Additional information

**Competing interests:** The authors declare no competing financial interests.

