## [Peer Review File · Nature Communications]

Reviewers' comments:

Reviewer #1 (Remarks to the Author):

Effector CD4⁺ T cells recognize intravascular antigen presented by patrolling monocytes. The authors use in vivo kidney imaging to investigate the role of intravascular antigen presentation in a T cell mediated model of glomerulonephritis in which a non-inflammatory IgG1 8D1-Ova fusion protein targets antigen to the glomerular basement membrane. They present data that CD4⁺ T cells adhere in glomerular capillaries in the steady-state and undergo increased intravascular migration during glomerular injury. They show that while MHCII was not expressed by glomerular endothelial cells, MHCII-expressing B cells and monocytes were retained in capillaries and could interact with CD4⁺ T cells. Following antigen challenge, CD4⁺ T cells increased their dwell time and intraluminal crawling behavior and showed increased expression of IFN γ when analyzed by ex vivo intracellular cytokine assays. Clodronate liposome-mediated depletion of monocytes inhibited injury, but B cells were shown to be dispensable, since μ MT KO mice developed glomerular injury in the model (neutrophil dwell time and ROS). The authors, propose a model where MHCII-expressing monocytes patrolling the glomerular capillaries acquiring antigen and then presenting this antigen to intravascular CD4⁺ T cells that contribute to neutrophil activation and glomerular injury. The primary strengths of the study are the focus on intravascular antigen presentation, which is interesting and understudied, the use of in vivo imaging to examine cell trafficking in the glomerular capillaries and the depletion approaches employed to assess the role of different MHCII populations in the model. The primary weaknesses of the study are a lack of an internal control for antigen specificity and some caveats associated with the hydronephrotic kidney imaging preparation. Also, it is unclear whether monocytes have adequate time in the glomeruli (tens of minutes in dwell time) to phagocytosis and present Ova antigen (typically hours in other systems).

Specific comments:

1. In the experiments using in vitro-activated adoptively-transferred T cells it would be best to co-transfer a second TCR transgenic T cell with irrelevant antigen specificity as an internal control for T cell retention, monocyte interactions and injury. Because many different cell types stick in the capillaries, retention and cell-cell interactions could be merely a consequence of occlusion or changes glomerular capillaries flow (i.e., due to arrested cells). How would you control for this possibility?
2. The study focuses on glomerular events, but antigen presentation could take place in other regions of the kidney leading to T cell activation and cytokine expression as measured in digested tissue. There is no direct evidence that T cell-monocyte interactions in the glomerular capillaries are actually the key antigen recognition event that leads to injury. Antigen is delivered systemically and could be captured by DCs in other regions of the kidney or even outside the kidney, so it possible that T cells recognize antigen in other places in the kidney or maybe distally in the spleen for example. Moreover, it seems that monocytes could captured antigen directly from the circulation. If T cells are activated outside the glomerulus, the T cell retention and glomerular injury observed may be a downstream consequence of antigen presentation occurring elsewhere.
3. Since the discovery of intravascular antigen presentation by monocytes is a new and exciting finding, would it be worth performing an experiment to show that the MHCII⁺ CD115⁺ cells can present 8D1-Ova to T cells in vitro?
4. How do speculate that T cells cause neutrophil activation in the short time they are retained in the glomerulus? Are you suggesting that IFN γ is released locally and rapidly to activate neutrophils? If cells are already activated in vitro then their cytokine responses may already be primed and the ex vivo assays more likely to reflect the in vitro stimulation rather than activation in the kidney. I realize the treatment with control 8D1 antibody doesn't show much IFN γ , but is that result expected? Would resting the T cells for 5-7 days before adoptive transfer change their trafficking behavior or cytokine expression?
5. Staining histological sections of normal adult kidneys and kidneys following 8D1-Ova treatment (with adoptively transferred T cells) could help verify the numbers of retained monocytes, B cell, T cells and neutrophils found in glomeruli and address some of the concerns that the hydronephrosis

imaging preparation leads to aberrant trafficking behavior.

6. Neutrophil sequestration in the lung microcirculation appears similar to the behavior in glomeruli with lots of abrupt cell sticking and discontinuous flowing.

7. Is there any concern that antibody labeling immune cells with antibodies in vivo alters their function or trafficking? Do naive dye-labeled T cells behave the same as antibody-labeled T cells in glomeruli?

8. Why would T cell recognition of antigen on monocytes decrease crawling speeds rather than induce transient arrest?

9. How is the intravascular antigen presentation mechanism seen in this model relevant to the induction of autoimmune response in the kidney? How would naïve T cells be activated in the first place, to generate activated effector cells that are retained and activated in the glomeruli to cause injury?

10. Glomerular dwell time is used throughout the manuscript as a proxy measure of cell activation or glomerular injury. But couldn't dwell time changes be due to multiple different factors including: changes in blood flow, vessel occlusion/vessel size restrictions, leukocyte integrin activation, T cell recognition of antigen. It would be helpful if this issue was discussed and taken into account when interpreting the data.

Reviewer #2 (Remarks to the Author):

This study uses multiphoton microscopy to analyse in vivo the interactions between selected circulating leukocytes populations and the capillary endothelium within glomeruli of normal mice; together with the physical and functional interactions between them. The study uses an approach developed and validated in the Hickey/Kitching Laboratory (Monash) that has previously been used to characterise patrolling neutrophils and monocytes within normal glomerular capillaries; and to show that patrolling monocytes were essential for T cell dependent neutrophil recruitment to the glomerulus (as cited in references 17 and 18 of the present manuscript). The current study the analysis to CD4 T cells to glomerular capillaries and their activation locally by intravascular antigen presenting cells. The experiments have been performed meticulously and the data are of very high quality. The new results extend knowledge and have clear implications for understanding immune mediated glomerular injury, and indeed for T cell responses to intravascular antigens more generally.

In essence, the results demonstrate that: (i) endogenous CD4 T cells labelled intravenously with anti-mouse CD4 antibodies adhere (defined as >30 seconds residence) transiently to normal glomerular endothelium and that 20% of adherent CD8 T cells display crawling behaviour; (ii) adoptively transferred fluorescently labelled OT-II T cells localise similarly to glomerular endothelium and that adhesion increases when the epitope they recognise (OVA 323-339) peptide is targeted to glomerular capillaries by conjugation to a monoclonal anti-GBM antibody (OVA/8D1), as evidenced by increases in both adhesion and the fraction of crawling CD4 cells; (iii) large numbers of EGFP-MHCII positive cells localise within glomerular capillaries but there is no evidence of extensions from DC in the surrounding interstitium in the glomeruli; (iv) there are abundant EGFP-MHCII positive cells within glomerular capillaries interact directly with labelled OTII cells with an average of 1 interaction of lasting around 4 minutes for each OTII cell, regardless of the local presence of OVA – OVA does however significantly increase the number of CD69 positive T cells and those expressing Th1 cytokines within the whole kidney; (v) around 80% of MHCII cells adherent to glomerular capillaries were B cells but these were dispensable for T cell dependent enhanced neutrophil adhesion; (vi) the remaining 20% of EGFP-MHCII positive cells were patrolling monocytes that could be identified by high expression of CD11b and had very prolonged dwell times in the glomerulus; and (vii) administration of clodronate loaded liposomes induced depletion of patrolling monocytes abrogated the T cell dependent enhanced neutrophil adhesion. From this the authors conclude that patrolling monocytes in the kidney present antigen to transiently infiltrating CD4 T cells within glomerular capillaries.

The data showing that CD4 T cells bind directly to normal glomerular capillaries and that crawling T cells interact directly with patrolling monocytes are highly convincing; as are those showing specific antigen (OVA 323-339) targeted to the glomerulus by conjugation to 8D1 - a monoclonal anti-GBM antibody – alters the OTII T cell behaviour and is essential for T cell dependent enhanced neutrophil infiltration. What for me remains an issue is whether the authors have proved that the antigen is presented by patrolling monocytes in the glomerulus or whether T cell activation could have occurred elsewhere with the glomerular effect reflecting (at least in part) the altered behaviour of already activated OTII cells infiltrating the glomeruli. I appreciate that the authors have provided indirect evidence to support their conclusions but I have four specific questions:

1. The 8D1/p323-339OVA is injected systemically together with OTII cells and so will be distributed systemically before being targeted to glomeruli by the 8D1 anti-GBM antibody. Do the authors know whether p323-339OVA conjugated to an irrelevant antibody does not have similar effects? The in vitro activation experiments are only a partial answer.
2. For obvious reasons, the demonstration of antigen specific significant increases in CD69 positive and IFN γ producing OTII T cells (Figure 4) were performed on whole kidney rather than glomerular T cells. Are these predominantly (or even exclusively) of glomerular origin, or is there also a T cell infiltrate in these post-hydronephrotic kidneys? If so it would be evidence of extra-glomerular but not necessarily extra-renal activation.
3. There was only a small time window when there were significantly more OTII cells in mice treated with 8D1/p323-339OVA than with 8D1 alone. Why should this be given that the OTII cells do not appear to transmigrate through the capillary wall and why does not antigen loaded patrolling monocytes not simply activate newly infiltrating OTII cells.
4. Presumably the p323-339OVA is targeted to the GBM on which the endothelium patrolled by the monocytes lie rather than remaining on the endothelial surface. If this is the case, it would be fascinating to know whether the monocytes also probe the baso-lateral surface of the endothelium. Do they?

Two final comments: the antigen used is a peptide potentially able to bind MHC class II on the cell surface without uptake or processing as would be required for whole antigen. This point should be noted in the discussion; the authors have made their text is unduly repetitive in their efforts to describe clearly exactly what each experiment consisted of. The increased stodginess more than outweighs increased enlightenment.

Reviewer #3 (Remarks to the Author):

This study has employed multi-photon intravital microscopy in a mouse model of glomerulonephritis. In this model animals are injected with a large number of ex vivo activated TH1-biased OT-II T cells and with a chimeric MAb, 8D1/pOVA, which targets a cognate antigenic peptide to the glomerular basement membrane. Previously published work in this experimental system, mainly by the Hickey group, has demonstrated that the ensuing glomerular inflammatory response depends on antigen recognition by CD4 T cells and that leukocytes undergo intravascular adhesion and crawling within glomerular capillaries. The present study attempts to address the question how the pathogenic CD4 T cells encounter cognate antigen. The authors propose that a subset of patrolling MHC-II+ monocytes in glomeruli can get access to deposited 8D1/pOVA and then present the OVA peptide to crawling OT-II effector T cells intravascularly, leading to T cell activation and subsequent neutrophil recruitment and ROS-mediated glomerulonephritis. Unfortunately, although the idea is intriguing, the conclusions raised by the authors are not adequately substantiated by the data presented. Specifically, the authors may consider the

following:

1. Evidence for moderate activation of OT-II cells in animals treated with 8D1/pOVA is provided in Fig. 4 where T cells were isolated from kidney digests and analyzed by FACS for expression of CD69 (Fig. 4d,e) and IFN γ (Fig. 4f,g). This effect of 8D1/pOVA on T cell activation phenotype was paralleled by a modest increase in the duration of interactions between OT-II cells and MHC-II+ cells crawling in glomerular capillaries. These correlative data are consistent with, but no rigorous evidence for the authors' contention that substantial T cell activation is achieved by intravascular antigen presentation in glomeruli. The authors did not address the possibility that circulating 8D1/pOVA was acquired, processed and presented to OT-II cells by APCs elsewhere, and OT-II cells migrated to the kidney in an already activated state. It is critical to examine the activation state of OT-II cells in other organs (spleen, blood, liver, bone marrow, lymph nodes) of 8D1/pOVA and control treated animals. What would be the effect if animals were treated with unconjugated MAb 8D1 and free OVA or pOVA conjugated to a MAb with different specificity?
2. Even if the activated OT-II cells were exclusively found in the kidney, this does not inevitably lead to the conclusion that activation occurred essentially within glomeruli. Multiple micrographs in the paper indicate that there is an abundance of MHC-II+ cells in extraglomerular renal tissue. How can the authors rule out that T cell activation did not occur in other parts of the kidney or by interactions with periglomerular MHC-II+ cells that may extend processes into the vessel lumen, as has been shown in the skin? Are peri-glomerular renal MHC-II+ cells susceptible to deletion by clodronate liposomes?
3. Intravital imaging studies of CD4 T cells in other tissues indicates that CD4 T cells, unlike CD8 T cells, require prolonged stable interactions with APCs (for several hours) to achieve an activated state. According to Fig. 4b and c, intravascular OT-II cells in glomeruli interacted, on average, with only one MHC-II+ cell for less than 5 minutes, and all contacts lasted less than 50 min. It is unclear whether such short contacts could be productive. It is conceivable that the ex vivo conditioning of OT-II cells prior to adoptive transfer may have lowered the requirement of T cells for extended antigen exposure, but this would have to be properly documented. Moreover, it needs to be shown that the observed T cell response is not merely an artifact of the non-physiological ex vivo conditioning protocol. Data with endogenously generated effector/memory cells need to be included.
4. No evidence is presented that the small fraction of MHC-II+ monocytes in peripheral blood are capable of capturing, processing and presenting antigen to CD4 T cells. It would be necessary to sort MHC-II+ monocytes from animals injected with pOVA/8D1 and perform an in vitro Ag presentation assay without any additional priming. Which costimulatory molecules and cytokines are involved in this process?
5. It is known that the CX3CR1+ patrolling monocytes express MHC-II and, in fact, previous studies have already reported a role for these monocytes in neutrophil retention and glomerulonephritis. The authors should check the expression of CX3CR1 in the MHC-II+ CD11b+ cells studied herein, since they might be the same population as previously studied.
6. Clodronate liposomes are a blunt tool to address the questions at hand. Systemic treatment with this reagent may not only deplete monocytes, but macrophages and dendritic cells in both intra- and (some) extravascular compartments. The effects observed with this treatment may not necessarily be attributed to monocytes. How does clodronate impact the adhesion and crawling of CD4 effector T cells?
7. To rule out the possibility that glomerulonephritis in 8D1/pOVA treated mice was not dependent on T cell activation, authors need to show that control animals injected with unconjugated 8D1 and OT-II cells do not develop glomerulonephritis. Based on the prior literature, the injection of the

antibody per se can be sufficient to produce monocyte-induced retention of neutrophils within glomeruli.

Effector CD4⁺ T cells recognize intravascular antigen presented by patrolling monocytes

The authors thank the reviewers for their supportive and positive comments regarding the initial version of the manuscript. We have addressed the reviewers' comments by performing numerous additional experiments. These experiments have improved the manuscript which has been extensively revised to accommodate our new work. These changes are described point-by-point below:

Reviewer #1:

Specific comments:

1. In the experiments using *in vitro*-activated adoptively-transferred T cells it would be best to co-transfer a second TCR transgenic T cell with irrelevant antigen specificity as an internal control for T cell retention, monocyte interactions and injury. Because many different cell types stick in the capillaries, retention and cell-cell interactions could be merely a consequence of occlusion or changes glomerular capillary flow (i.e., due to arrested cells). How would you control for this possibility?

To address this point we performed the suggested experiment, co-transferring CD4⁺ T cells from the SMARTA-GFP mice as a second TCR transgenic mouse of irrelevant antigenic specificity. The CD4 T cells in these mice recognise the LCMV GP-derived P13 peptide, and also express GFP. SMARTA T cells underwent the same *in vitro* activation protocol as the OT-II (but using the LCMV peptide instead of the OVA peptide). SMARTA cells achieved a comparable level of activation as routinely observed with OT-II cells, determined by flow cytometric assessment of IFN γ expression. The activated SMARTA cells were then co-transferred with OT-II cells into recipient mice treated with 8D1-pOVA, i.e. the target peptide for OT-II cells. SMARTA cells were identified on the basis of GFP expression while OT-II cells were labelled with CMTPX and the behaviour of the two types of cells was assessed in the same glomeruli. These side-by-side analyses revealed that the retention time of the OT-II cells was significantly longer than that of the SMARTA T cells (**Fig. 1** below). Importantly, this difference was not seen in the absence of antigen, i.e. in mice receiving 8D1 alone (data not shown). These findings validate the concept that the prolonged glomerular retention of OT-II cells in mice treated with 8D1-pOVA stemmed from antigen recognition by these cells, rather than other factors such as capillary occlusion and changes in blood flow that would apply to both types of cells. These new data have been added to an updated **Figure 2** in the revised manuscript.

Figure 1: Increased glomerular retention of OT-II cells in presence of 8D1-pOVA requires the appropriate TCR. Activated OT-II (pOVA-specific) and SMARTA (LCMV-specific) activated CD4⁺ T cells were co-transferred into 8D1-pOVA-treated mice, and glomerular retention of the two types of cells was assessed using multiphoton microscopy. SMARTA cells were identified on the basis of GFP expression while OT-II cells were labelled with CMTPX. Shown are individual T cell dwell times 30-120 min after cell transfer, as well as group mean \pm sem. Data represent analysis of a total 139 (SMARTA) and 127 (OT-II) cells from n=3 recipient mice. *, p > 0.05.

2. The study focuses on glomerular events, but antigen presentation could take place in other regions of the kidney leading to T cell activation and cytokine expression as measured in digested tissue. There is no direct evidence that T cell-monocyte interactions in the glomerular capillaries are actually the key antigen recognition event that leads to injury. ... If T cells are activated outside the glomerulus, the T cell retention and glomerular injury observed may be a downstream consequence of antigen presentation occurring elsewhere.

Response: In our original manuscript, it was not unreasonable for the reviewer to contend that recognition of antigen outside the glomerulus might have been responsible for events inside the glomerulus. In summary, we have now performed additional experiments (detailed below) that demonstrate that this is not the case. The first two experiments involved providing OVA-peptide to recipient mice using different targeting strategies. These experiments demonstrated that antigen present elsewhere in the body did not induce the glomerular T cell and neutrophil events that we observe when antigen is present in the glomerulus. The third experiment used an NFAT-GFP reporter system in OT-II cells as a marker for T cell signalling in response to antigen. The studies show nuclear translocation of NFAT in T cells in glomeruli in response to antigen, demonstrating direct activation of effector T cells by antigen in glomeruli.

Firstly, to address the question of extraglomerular T cell activation resulting in non-specific glomerular changes, we performed experiments comparing T cell responses in the kidney when antigenic peptide was either targeted to the glomerulus by conjugation to 8D1 (8D1/pOVA, as in our original experiments) or non-specifically targeted via conjugation of pOVA to a non-targeted isotype control antibody (IgG/pOVA). We first examined OT-II T cell activation *in vivo*, in the presence of these forms of antigen, using flow cytometry to assess T cell IFN γ production as we had previously in Figure 4. The findings demonstrated that OT-II cells undergo activation in the kidney when pOVA is conjugated to 8D1, but do not when pOVA is bound to a non-targeted antibody (Fig. 2 below). These new data have been included in a new Figure 5 in the revised manuscript (as Figure 5a-c).

Figure 2: Specific targeting of pOVA to the glomerulus is required for T cell activation in the kidney. Assessment of OT-II cell activation in the kidney in the presence of glomerulus-targeting antibody alone (8D1), glomerulus-targeted antigenic peptide (8D1/pOVA), or non-targeted antigenic peptide (IgG/pOVA). **A, B:** Example flow cytometry data showing gating strategy to detect OT-II cells (CD4⁺ V α 2⁺) in kidney digests (A), followed by intracellular staining for IFN γ in all three experimental conditions (B). **C:** Group data showing IFN γ expression (as % of OT-II cells) from mice receiving either 8D1, 8D1/pOVA or IgG/pOVA along with OT-II cells. Data are shown as mean \pm sem of n=7 (8D1) or 8 (8D1/pOVA & IgG/pOVA) mice/group, from 5 individual experiments. ***, P<0.001; ****, P<0.0001

Secondly, addressing the same question, we used intravital microscopy to compare the response of neutrophils in the glomerulus in mice receiving OT-II cells plus either 8D1/pOVA or IgG/pOVA, examining neutrophil behaviour at 24 h, as we had done previously in **Figure 7**. These data show that in mice that received non-targeted pOVA, neutrophil retention and ROS production were not different from that seen in mice receiving unconjugated antibody, showing minimal evidence of the local activation seen in mice receiving the glomerulus-targeted 8D1/pOVA conjugate (**Fig. 3** below). These new findings provide further support that the localisation of the antigen to the glomerulus is key for the induction of pro-inflammatory changes within the glomerulus. These new data have been added to **Figure 5** of the revised manuscript, as **Figure 5 c & d**.

Figure 3: Specific targeting of pOVA to the glomerulus is required for neutrophil activation in the kidney. Neutrophil behaviour in the glomerulus, in the presence of glomerulus-targeting antibody alone (8D1), glomerulus-targeted antigenic peptide (8D1/pOVA), or non-targeted antigenic peptide (IgG/pOVA), as assessed by multiphoton microscopy. Data are shown for static dwell time (**A**) and % of adherent neutrophils that were DHE⁺ (**B**). Data are shown as mean \pm sem of $n=8$ (8D1 & IgG/pOVA) or 10 (8D1/pOVA) mice/group. *, $P<0.05$; **, $P<0.01$ via Kruskal-Wallis test with multiple comparisons.

Thirdly, we performed experiments to directly assess T cell activation in the glomerulus using a system enabling visualisation of translocation of the transcription factor nuclear factor of activated T cells-1 (NFAT1) to the nucleus (1, 2). NFAT1, a critical transcription factor in T cells driving changes in gene expression in response to T cell activation, rapidly translocates to the nucleus in response to elevation in cytosolic calcium following TCR engagement (3). Previous *in vivo* imaging studies have used a GFP-tagged version of NFAT1 (“NFAT-GFP”) to track T cell activation *in vivo* using multiphoton microscopy (1, 2). We implemented this system in our model by generating NFAT-GFP-transduced OT-II (OT-II_{NFAT-GFP}) cells. We first demonstrated *in vitro* that the NFAT-GFP reporter readily translocated to the nucleus following T cell activation and underwent *in vitro* activation in a similar manner as non-transduced OT-II cells (shown below in **Fig. 4** below, and included in the revised manuscript as **Suppl. Fig. 4**).

We then transferred these cells into mice 30 minutes after the mice were treated with either 8D1 or 8D1/pOVA and assessed subcellular localization of the reporter. In 8D1-pOVA-treated mice, 26% of OT-II_{NFAT-GFP} cells displayed translocated NFAT-GFP, whereas in mice that received 8D1, translocation was seen in only 10% (*, $P < 0.05$ via Fisher’s exact test) (see **Fig. 5** below, and now included in the revised manuscript as part of a revised **Figure 4 (panels d-g)** and in the new **Suppl. Video S8**). Furthermore, typically T cells with cytoplasmic NFAT-GFP were highly migratory, while cells displaying nuclear NFAT-GFP remained static (new **Suppl. Video S8**), consistent with previous descriptions of T cell arrest during antigen-dependent activation (4). These experiments provide direct evidence of antigen-dependent T cell activation occurring within the glomerular capillaries.

Collectively these findings demonstrate that targeting of antigen to glomerulus via 8D1 is necessary for the complete T cell and neutrophil response in the kidney, and that T cells do undergo activation in the glomerulus following their recruitment into this vascular bed.

Figure 4

Figure 4: In vitro-activated OT-II_{NFAT-GFP} cells display nuclear NFAT-GFP translocation and IFN γ production. OT-II_{NFAT-GFP} cells were stained with CMTPX (red) and examined via live cell confocal microscopy following activation by PMA/Ionomycin (a – time after activation shown below images). The images show the change in sub-cellular localization of NFAT-GFP (green) from cytoplasmic (80 s), to both nuclear and cytoplasmic (220 s), to exclusively nuclear (600 s). (b) Comparison of IFN γ production by NFAT-GFP-transduced OT-II cells with non-transduced OT-II cells from the same culture. OT-II cells, identified on the basis of CD4 (not shown) and V α 2 expression were separated into transduced and non-transduced cells on the basis of GFP expression, and IFN γ production following in vitro activation assessed in both populations, via intracellular staining. The percentage of cells positive for IFN γ was comparable for both populations.

Figure 5

Figure 5: Use of OT-II_{NFAT-GFP} cells to demonstrate antigen-dependent T cell activation in glomerular capillaries. (a-d): Images of OT-II_{NFAT-GFP} cells in vivo in glomerular capillaries, illustrating OT-II cells with NFAT-GFP in cytoplasmic (a), nuclear and cytoplasmic (b) and exclusively nuclear (c) locations. The cytoplasm is visible via CMTPX staining (red) while NFAT-GFP is visible in green. Scale bars denote 10 μ m. (d): Graphs showing percentages of T cells with NFAT-GFP in cytoplasmic, nuclear and cytoplasmic and nuclear locations, shown for mice treated with either 8D1 or 8D1/pOVA. Data show analysis of 51 cells from 3 independent experiments (8D1) and 89 cells from 4 independent experiments (8D1/pOVA). *, $P < 0.05$ via Fisher's exact test, comparing the number of cells with cytoplasmic NFAT versus the combined total of the nuclear/cytoplasmic and nuclear categories in the two treatment groups.

In addition to these new and important experiments, the text of the manuscript has also been clarified with regard to the steps in antigen-mediated T cell activation and what elements of this are being examined in this study. To this end we have included this new section in the Introduction:

“Glomerular injury mediated by effector CD4⁺ T cells in of glomerulonephritis involves multiple steps, beginning with loss of tolerance to nephritogenic autoantigens in secondary lymphoid organs.(5, 6) This hypothesis is supported by the discovery of circulating CD4⁺ T cells specific for these antigens in patients with autoimmune glomerulonephritis.(7-9) In patients with autoimmune disease, circulating autoreactive T cells preferentially display a memory phenotype, indicating that they have been exposed to cognate antigen and undergone differentiation into effector or memory T cells.(10, 11) However, the existence of circulating, antigen-experienced T cells is insufficient to result in disease. The final steps in the process involve T cell recognition of antigen in the target tissue, leading to effector T cell-mediated injury at the site of antigen recognition, the glomerulus. Indeed, analysis of responses of antigen-experienced T cells to antigen in the periphery has shown that these cells can respond within minutes upon recognition of cognate antigen.(4, 12)”

As the reviewer knows, current paradigms relating to T cell recognition, based on experimental data derived over many years, dictate that naïve T cells encounter antigen in secondary lymphoid organs where they differentiate and acquire effector function. As we are defining antigen *recognition* in the glomerulus, a peripheral site, our studies were designed to bypass these initial steps by which naïve T cells differentiate in response to primary antigen presentation. In most forms of glomerulonephritis, antigen is available systemically, but antigen-specific CD4⁺ effector responses in peripheral tissues require local antigen recognition. Our revised manuscript now more clearly demonstrates this antigen-specific recognition by effector CD4⁺ T cells in glomeruli, via the NFAT-GFP experiments described above. These observations have been described in the Results section of the revised manuscript as follows:

“To assess whether OT-II cells undergo activation within the glomerular capillaries, we made use of a system enabling visualization of translocation of the transcription factor nuclear factor of activated T cells-1 (NFAT1) to the nucleus.(1, 2) In T cells, NFAT is a critical transcription factor driving changes in gene expression in response to T cell activation, and rapidly translocates to the nucleus in response to elevation in cytosolic calcium following TCR engagement.(3) Via use of the fluorescent reporter NFAT1₍₁₋₄₆₀₎-GFP (“NFAT-GFP”), NFAT1 translocation to the nucleus, and therefore T cell activation, can be tracked in vivo using multiphoton microscopy. We generated NFAT-GFP-transduced OT-II (OT-II_{NFAT-GFP}) cells and demonstrated in vitro that the NFAT-GFP reporter readily translocated to the nucleus following T cell activation (**Suppl. Fig. 4**). We then transferred these cells into mice 30 min after treatment with either 8D1 or 8D1-pOVA and assessed subcellular localization of the reporter (**Fig. 4d-f & Suppl. Fig. 4c-e**). Following administration of 8D1-pOVA, NFAT-GFP had either partially or completely translocated to the nucleus in 26% of OT-II_{NFAT-GFP} cells, whereas in mice that received 8D1, translocation was seen in only 10% of cells ($P < 0.05$, Fisher’s exact test) (**Fig. 4g**). Furthermore, typically T cells with cytoplasmic NFAT-GFP were active and highly migratory, while cells displaying nuclear NFAT-GFP remained static (**Suppl. Video S8**), consistent with previous descriptions of T cell arrest during antigen-dependent activation.(4) These experiments provide direct evidence of T cell activation occurring within the glomerular capillaries.”

Finally we have included a new section in the Discussion addressing the implications of these new findings, as follows:

“We hypothesized that a key step in the development of T cell mediated glomerulonephritis takes place within the glomerular microvasculature, when T cells recognize disease-causing antigens in glomerular capillaries. Here we provide direct evidence of T cell activation in the glomerulus during the initial response to locally planted antigen. Using the NFAT-GFP reporter to reveal T cell activation *in vivo* at a single cell level, we directly visualized increased nuclear translocation of this transcription factor in T cells within the glomerular capillaries, in the presence of cognate antigen targeted to the glomerulus. While it is possible that antigen presentation to T cells takes place in other regions of the kidney, these findings clearly demonstrate that antigen-dependent T cell activation does occur intravascularly within the glomerular capillaries. This contention is strengthened by findings that glomerular targeting of pOVA was necessary for both T cell activation and downstream neutrophil activation, in that these responses were not seen when pOVA was delivered systemically in a non-targeted fashion. Together these data provide compelling evidence that T cells can respond to peptide antigen within the glomerular capillaries and that antigen recognition in this location can lead to proinflammatory changes in the glomerular microvasculature.”

3. Since the discovery of intravascular antigen presentation by monocytes is a new and exciting finding, would it be worth performing an experiment to show that the MHCII⁺ CD115⁺ cells can present 8D1-OVA to T cells *in vitro*?

Response: Thank you for this helpful suggestion. We have now performed *in vitro* experiments to demonstrate that MHCII⁺ CD115⁺ monocytes can present antigen. We sorted CD115⁺ MHCII⁺ monocytes from the blood of MHCII-GFP mice (excluding CD19⁺ B cells), and used this population in OT-II cell proliferation assays, as we have done with other APC populations in the past (13, 14). These experiments revealed that MHCII⁺ monocytes have the capacity to induce OT-II T cell proliferation under conditions where antigenic OVA-peptide is present (Fig. 6 below). In these experiments, T cells did not proliferate in the absence of APCs and the degree of T cell proliferation was proportional to the number of monocytes in the assay (Fig. 6 below). Splenocytes were used a positive control for induction of T cell proliferation. These observations indicate that the MHCII-expressing subset of circulating monocytes has the capacity to present antigen to CD4⁺ T cells and induce their proliferation. These new data have been included as a new Figure 7 in the revised manuscript.

Figure 6: MHCII⁺ monocytes are capable of inducing CD4⁺ T cell proliferation. MHCII⁺ monocytes were sorted from the blood of MHCII-GFP mice and used as antigen-presenting cells in *in vitro* CFSE dilution T cell proliferation assays with OT-II cells. **A:** Flow cytometry plots showing gating strategy used to isolate MHCII⁺ monocytes, initially gating on forward and side scatter profiles (left panel), then identifying monocytes (CD19⁻ CD115⁺ - central panel), then selecting GFP⁺ cells within the monocyte population (right hand panel). **B-D:** OT-II cell proliferation as measured via CFSE dilution. **B:** Representative FACS plots of proliferating OVA-specific Vα2⁺ OT-II CD4⁺ T cells shown for T cells cultured with peptide-pulsed MHCII⁺ monocytes versus T cells with peptide alone. CFSE dilution (proliferation) is observed in the presence of MHCII⁺ monocytes whereas no proliferation is observed in the absence of monocytes. Inset numbers represent

*the average proportion of cell populations over 5 individual experiments. C, D: Group data from individual experiments performed with T cells and OVA peptide alone, or with T cells, OVA peptide and MHCII⁺ monocytes, showing OT-II cell proliferation as % (c) and number (d) of OT-II cells. *, P < 0.05 versus T cells and peptide alone.*

4. How do you speculate that T cells cause neutrophil activation in the short time they are retained in the glomerulus? Are you suggesting that IFN γ is released locally and rapidly to activate neutrophils? If cells are already activated in vitro then their cytokine responses may already be primed and the ex vivo assays more likely to reflect the in vitro stimulation rather than activation in the kidney. I realize the treatment with control 8D1 antibody doesn't show much IFN γ , but is that result expected? Would resting the T cells for 5-7 days before adoptive transfer change their trafficking behavior or cytokine expression?

Response: Our past experience with this OT-II cell model has shown that even though the OT-II cells have been differentiated *in vitro* and are of the effector phenotype, following their *in vivo* transfer they require the presence of antigen in the host to generate detectable IFN γ . Indeed we confirmed these findings in the course of these revisions, observing that following their transfer into a recipient mouse, T cells only generated significant amounts of IFN γ when exogenous pOVA was administered (see **Fig. 2b, c** above). This is consistent with the immunological concept that naïve T cells first undergo activation via recognition of their antigen to proliferate, differentiate and transition to an effector or memory phenotype, but must subsequently detect antigen again in the periphery to provide an effector response (5).

The mechanisms whereby T cells induce neutrophil retention and activation in glomeruli following local recognition of antigen are currently unknown but could include cell contact-dependent mechanisms and release of soluble factors. In terms of timing, changes in glomerular neutrophil behaviour can occur within minutes of stimulation (15, 16) and effector T cells can rapidly respond to antigen in other tissues (4, 17). Thus the time frame of our observations is consistent with previous data. Further investigation of this issue will form the basis of future studies.

5. Staining histological sections of normal adult kidneys and kidneys following 8D1-Ova treatment (with adoptively transferred T cells) could help verify the numbers of retained monocytes, B cell, T cells and neutrophils found in glomeruli and address some of the concerns that the hydronephrosis imaging preparation leads to aberrant trafficking behavior.

Response: The 4-dimensional (3D plus time) imaging technique that we use to assess whole glomeruli over time is a more sensitive and informative technique for detection of intraglomerular leukocytes than 2D immunohistological analyses. Conventional histology takes only a ~4 μ m section of the glomerulus at a single time point post mortem. As such we feel there is limited value in these experiments. We have replicated observations re intraglomerular trafficking of MHCII⁺ cells in hydronephrotic kidneys with similar data from intact kidneys.

6. Neutrophil sequestration in the lung microcirculation appears similar to the behavior in glomeruli with lots of abrupt cell sticking and discontinuous flowing.

Response: The reviewer is correct in their comment: based on the results of recent studies using intravital microscopy to examine neutrophil dynamics in the lung, there is some similarity between neutrophil behaviour in capillaries in the lung to that in the glomerulus (18-20). In the lung, neutrophils adopt either static or crawling phenotypes, in roughly equal numbers. We have

observed similar behaviours in the glomerular capillaries previously (15, 16), and in the present study. We have added a section to the Discussion to comment on this, as follows:

“The behaviours of neutrophils observed in glomerular capillaries under resting conditions are similar to those recently reported for neutrophils in capillaries of the uninflamed lung. In the lung, these behaviours may be a form of immune surveillance facilitating rapid responses to local infection. While it is possible that this is also an important function of neutrophils in glomeruli, microbial infection is less explicitly relevant in the glomerular microcirculation than in the lung. Moreover, in the glomerulus this behaviour may be a double-edged sword in that it also underlies the induction of injurious responses after immune complex deposition or other inflammatory stimuli.”

7. Is there any concern that antibody labeling immune cells with antibodies *in vivo* alters their function or trafficking? Do naive dye-labeled T cells behave the same as antibody-labeled T cells in glomeruli?

Response: We have addressed this question in previous studies, most notably in Devi *et al* (2013) (15) in which we compared glomerular neutrophil trafficking as detected using anti-Gr-1 and in LysM-eGFP neutrophil reporter mice, seeing no differences in neutrophil behaviour assessed using these approaches. To further address the present comments, we have now performed additional experiments using OT-II cells. We performed side-by-side experiments comparing the intraglomerular behaviour of OT-II cells labelled with either CFSE, a well-characterised intracellular dye commonly used in trafficking studies, or anti-CD45 (1 µg/mL, similar to the antibody concentration achieved in these studies when administering into the circulation). These experiments revealed no significant differences in the number or dwell time of OT-II cells retained within glomerular capillaries (**Fig. 7** below). As a result of these findings and our previous experience with other immune cell subsets, we are confident that antibody labelling has minimal impact on immune cell trafficking in the glomerulus.

Figure 7: Comparison of OT-II cell trafficking parameters for cells labelled with either CFSE or extracellular antibody. OT-II T cells underwent the standard activation protocol and were subsequently labelled with either CFSE or PE-conjugated anti-CD45 (1 µg/mL, 15 min), before being co-transferred into mice undergoing renal intravital

microscopy. OT-II cell adhesion (A, shown for total, static and crawling cells) and dwell time (B, shown for total cells) were examined for T cells in the same glomeruli. Data are shown as mean ± sem of n=6 individual experiments, representing analysis of 125 (CFSE) and 134 (anti-CD45) cells/group.

8. Why would T cell recognition of antigen on monocytes decrease crawling speeds rather than induce transient arrest?

Response: The expectation of seeing arrest of T cells during antigen presentation stems from previous observations of archetypal forms of antigen presentation that occur in lymph nodes or

peripheral tissues, i.e. outside the vasculature. While this may be the case within the vasculature, in interpreting the result from the present manuscript, it is important to recognise that the OT-II migration speed data are derived from all crawling T cells. The T cells that are assessed are not restricted to those interacting with monocytes, as these experiments were not performed in MHCII-GFP reporter mice. As such they represent average data from all T cells observed to undergo retention in the glomerulus, not only those that encountered antigen-presenting cells. However, we have now been able to unequivocally identify T cells undergoing antigen-dependent activation within the glomerulus, via assessment of NFAT nuclear translocation. In these experiments, all cells with nuclear NFAT, i.e. those undergoing antigen recognition, were immobile, while non-activated cells with cytoplasmic NFAT were typically highly migratory. This increased level of sensitivity for detection of activated cells has therefore allowed us to generate data consistent with previous observations of T cell arrest during antigen recognition. This new observation is described in the revised text in the section discussing the OT-II NFAT-GFP cells, as follows:

“Furthermore, typically T cells with cytoplasmic NFAT-GFP were active and highly migratory, while cells displaying nuclear NFAT-GFP remained static (Suppl. Video S8), consistent with previous descriptions of T cell arrest during antigen-dependent activation.(4)”

and is illustrated in a new video (Video S8) included in the revised manuscript

9. How is the intravascular antigen presentation mechanism seen in this model relevant to the induction of autoimmune response in the kidney? How would naïve T cells be activated in the first place, to generate activated effector cells that are retained and activated in the glomeruli to cause injury?

Response: We are studying antigen recognition in the glomerulus by effector T cells, not loss of tolerance. As discussed in the response to Comment #2, the pathogenesis of T cell-mediated glomerulonephritis is a multi-step process, involving loss of tolerance to nephritogenic antigens, T cell differentiation in secondary lymphoid organs and then T cell recognition of antigen in the glomerulus. In the majority of forms of autoimmune glomerulonephritis, antigen is systemic. Loss of tolerance is systemic, while glomerular injury results from local antigen-specific recognition by effector T cells. The aspect of the response specifically focussed on in this study is the latter step, i.e. the response of antigen-experienced T cells to detection of cognate antigen in the glomerulus.

It is clear from this question that we have not clearly highlighted the aspect of this process that we are examining in this study. To improve the clarity of the manuscript, in the revised version we have added the following passage to the Introduction:

“Glomerular injury mediated by effector CD4⁺ T cells in of glomerulonephritis involves multiple steps, beginning with loss of tolerance to nephritogenic autoantigens in secondary lymphoid organs.(5, 6) This hypothesis is supported by the discovery of circulating CD4⁺ T cells specific for these antigens in patients with autoimmune glomerulonephritis.(7-9) In patients with autoimmune disease, circulating autoreactive T cells preferentially display a memory phenotype, indicating that they have been exposed to cognate antigen and undergone differentiation into effector or memory T cells.(10, 11) However, the existence of circulating, antigen-experienced T cells is insufficient to result in disease. The final steps in the process involve T cell recognition of antigen in the target tissue, leading to effector T cell mediated injury at the site of antigen recognition, the glomerulus. Indeed, analysis of responses of antigen-experienced T cells to antigen in the periphery has shown that these cells can respond within minutes upon recognition of cognate antigen.(4, 12) However, the mechanism whereby CD4⁺ T cells recognise antigens in the unique microvasculature of the glomerulus is not known.”

10. Glomerular dwell time is used throughout the manuscript as a proxy measure of cell activation or glomerular injury. But couldn't dwell time changes be due to multiple different factors including: changes in blood flow, vessel occlusion/vessel size restrictions, leukocyte integrin activation, T cell recognition of antigen. It would be helpful if this issue was discussed and taken into account when interpreting the data.

Response: While it is possible that some of the changes in immune cell dwell time seen in the glomerulus stem from factors including vascular occlusion and alterations in blood flow, we now have shown using T cells from SMARTA mice (see response to Comment #1), that these potential other elements are not the reason for the antigen-specific changes that we see. To further address this point, we have added the following section to the Discussion:

"We used dwell time as a further readout of immune cell activation in the glomerulus. In previous studies this parameter has correlated well with neutrophil activation and glomerular injury and is adhesion molecule-dependent (15, 16). In addition, here we show that T cells incapable of recognizing antigen in the glomerulus do not undergo the increased dwell time shown by OVA-specific T cells, demonstrating a role for antigen recognition in this response. Despite these observations, it should also be noted that other factors potentially at play in the inflamed glomerulus, such as reduced blood flow, may exert further influence on immune cell dwell time (21).

Reviewer #2:

What for me remains an issue is whether the authors have proved that the antigen is presented by patrolling monocytes in the glomerulus or whether T cell activation could have occurred elsewhere with the glomerular effect reflecting (at least in part) the altered behaviour of already activated OTII cells infiltrating the glomeruli. I appreciate that the authors have provided indirect evidence to support their conclusions

Response: We have performed significant new experiments to exclude this as a major concern, detailed below:

- 1) OVA peptide-IgG (i.e. antigen provided to the recipient that was not targeted to the glomerulus) did not result in changes in glomerular T cell behaviour nor the accumulation of IFN γ -secreting T cells in the kidney (see response to Comment #1 below).
- 2) Using an NFAT-GFP reporter system in OT-II cells, we observed nuclear translocation of NFAT *in vivo* in T cells in glomeruli (detailed in response to **Reviewer #1, Comment #2**). This occurs in effector T cells when MHCII-antigen complexes interact with the TCR. We have therefore directly demonstrated antigen-specific signalling that leads to activation within the glomerulus *in vivo*.

*1. The 8D1/p323-339OVA is injected systemically together with OTII cells and so will be distributed systemically before being targeted to glomeruli by the 8D1 anti-GBM antibody. Do the authors know whether p323-339OVA conjugated to an irrelevant antibody does not have similar effects? The *in vitro* activation experiments are only a partial answer.*

Response: To address this point, we performed additional experiments to compare T cell responses in the kidney when antigenic peptide was either targeted to the glomerulus by conjugation to 8D1 (8D1/pOVA) or non-specifically targeted via conjugation of pOVA to a non-targeted isotype control antibody (IgG/pOVA). We first examined OT-II T cell activation *in vivo*, in the presence of these forms of antigen, using flow cytometry to assess T cell IFN γ production as we had previously in Figure 4. The findings demonstrated that OT-II cells undergo activation in the kidney when pOVA is conjugated to 8D1, but do not when pOVA is bound to a non-targeted antibody (see **Fig. 2** above, and included in the revised manuscript as **Figure 5**).

Secondly, we used intravital microscopy to compare the response of neutrophils in the glomerulus in mice receiving OT-II cells plus either 8D1/pOVA or IgG/pOVA, examining neutrophil retention and ROS production at 24 h, as we had done previously in Figure 7. These data show that neutrophil behaviour in mice that received non-targeted pOVA was not different from that in mice receiving unconjugated antibody, showing minimal evidence of the activation seen in mice receiving the glomerulus-targeted 8D1/pOVA conjugate (see **Fig. 3** above).

These new findings, now included in the manuscript, provide further support that the localisation of the antigen to the glomerulus is key for the induction of pro-inflammatory changes within the glomerulus.

2. For obvious reasons, the demonstration of antigen specific significant increases in CD69 positive and IFN γ producing OTII T cells (Figure 4) were performed on whole kidney rather than glomerular T cells. Are these predominantly (or even exclusively) of glomerular origin, or is there also a T cell infiltrate in these post-hydronephrotic kidneys? If so it would be evidence of extra-glomerular but not necessarily extra-renal activation.

Response: Firstly, it is important to point out that these experiments were performed in intact rather than post-hydronephrotic kidneys, although it is correct to state that these experiments do

not differentiate between glomerular and interstitial T cells. However, as part of these revisions we have now used *in vivo* visualisation of NFAT nuclear translocation to directly demonstrate OT-II cell activation in the glomerulus (see details in the response to Reviewer 1, Comment #2, and **Figure 4** and **Supplementary Figure 4** in the revised manuscript). These new experiments clearly demonstrate that these cells undergo antigen-dependent activation within the glomerular capillaries.

3. There was only a small time window when there were significantly more OTII cells in mice treated with 8D1/p323-339OVA than with 8D1 alone. Why should this be given that the OTII cells do not appear to transmigrate through the capillary wall and why does not antigen loaded patrolling monocytes not simply activate newly infiltrating OTII cells.

Response: The reason for this brief time window of antigen-specific T cell activation is unclear, but we are reassured that our observations are consistent with the literature in other peripheral sites. For example, Honda *et al.* showed that the reduction in T cell migration seen rapidly after antigen recognition dissipated in the hours after antigen was administered (4). In a similar vein, Hwang *et al.* demonstrated that if T cells were not allowed to enter a site of antigen presentation until 2 hours after antigen application, their capacity to induce a T cell-dependent response was significantly attenuated (17). These studies demonstrate that complexities exist in antigen presentation and the manner in which T cells respond at different times following antigen administration.

In regards to the present study, one possibility is that there is an initial peak of antigen presentation in the period immediately following antigen deposition in the glomerulus, and that after this period the rate of antigen presentation is reduced and difficult to detect using the approaches employed here. One could also speculate that the induction of inflammation resulting from initial antigen presentation alters the function and behaviour of immune cells, particularly monocytes in the glomerulus, and that OT-II cells that enter the glomerulus after this stage do not receive signals of the same nature as they do prior to induction of inflammation.

4. Presumably the p323-339OVA is targeted to the GBM on which the endothelium patrolled by the monocytes lie rather than remaining on the endothelial surface. If this is the case, it would be fascinating to know whether the monocytes also probe the baso-lateral surface of the endothelium. Do they?

Response: While multiphoton imaging does not have sufficiently high resolution to address this issue, we examined this issue by performing transmission electron microscopy (TEM) of glomeruli, to investigate intravascular leukocytes. Via this approach, leukocyte projections could be detected in close proximity to endothelial fenestrae in glomerular capillaries (**Fig. 8** below). These observations suggest that these leukocytes can actively probe these locations. However, a definitive answer for this point requires dynamic *in vivo* imaging experiments at higher resolution than is currently possible. Nevertheless, as these new electron microscopy observations do contribute to our mechanistic understanding they have been included in the Results section of the revised manuscript as supplementary data (see **Suppl. Fig. 8**).

Figure 8: Electron microscopic assessment of immune cell probing adjacent to endothelial fenestrations in glomerular capillaries. Otherwise untreated kidneys were fixed and prepared for electron microscopy. Images show mononuclear leukocytes within glomerular capillaries, closely apposed to the endothelial surface. Regions within insets (magnified in the top right corner) show leukocyte microvilli within (arrow) or adjacent to fenestrations. Scale bars indicate either 0.2 or 0.5 μm .

5. Two final comments: the antigen used is a peptide potentially able to bind MHC class II on the cell surface without uptake or processing as would be required for whole antigen. This point should be noted in the discussion; the authors have made their text is unduly repetitive in their efforts to describe clearly exactly what each experiment consisted of. The increased stodginess more than outweighs increased enlightenment.

Response: To address the first comment, we have the following passage to the Discussion examining the issue of direct peptide presentation:

“In some studies, immunogenic peptides capable of binding directly to MHCII without intracellular uptake or processing have been used to examine the immediate response to antigen presentation (4). As our studies involve use of the peptide recognised by OT-II cells, it is conceivable that a similar response could be at work here. However, as the peptide is covalently conjugated to a much larger immunoglobulin molecule, we anticipate that intracellular processing is the more likely route by which the OVA peptide is loaded onto MHCII.”

To address the second comment, we have also re-examined and edited the Results and Discussion sections with a view to reducing repetition and stodginess, removing over 1000 words in the process, including several paragraphs from the original Discussion. Thank you for this suggestion – we feel the manuscript is now more concise and clear.

Reviewer #3:

1.1 The authors did not address the possibility that circulating 8D1/pOVA was acquired, processed and presented to OT-II cells by APCs elsewhere, and OT-II cells migrated to the kidney in an already activated state... What would be the effect if animals were treated with unconjugated MAb 8D1 and free OVA or pOVA conjugated to a MAb with different specificity?

Response: To address the question of extraglomerular T cell activation resulting in non-specific glomerular changes, we elected to take the second approach suggested here, of comparing responses to 8D1/pOVA to those to pOVA conjugated to a different mAb. We compared T cell responses in the kidney when antigenic peptide was either targeted to the glomerulus by conjugation to 8D1 (8D1/pOVA, as in our original experiments) or non-specifically targeted via conjugation of pOVA to a non-targeted isotype control antibody (IgG/pOVA). We first examined OT-II T cell activation *in vivo*, in the presence of these forms of antigen, using flow cytometry to assess T cell IFN γ production as we had previously in Figure 4. The findings demonstrated that OT-II cells undergo activation in the kidney when pOVA is conjugated to 8D1, but do not when pOVA is bound to a non-targeted antibody (**Fig. 2** above). These new data have been included in a new **Figure 5** in the revised manuscript as **Figure 5a-c**.

Additionally, we used intravital microscopy to compare the response of neutrophils in the glomerulus in mice receiving OT-II cells plus either 8D1/pOVA or IgG/pOVA, examining neutrophil behaviour at 24 h, as we had done previously in **Figure 7**. These data show that in mice that received non-targeted pOVA, neutrophil retention and ROS production were not different from that seen in mice receiving unconjugated antibody, showing minimal evidence of the local activation seen in mice receiving the glomerulus-targeted 8D1/pOVA conjugate (**Fig. 3** above). These new findings provide further support that the localisation of the antigen to the glomerulus is key for the induction of pro-inflammatory changes within the glomerulus. These new data have been added to **Figure 5** (panels **c** and **d**) in the revised manuscript.

As an alternative approach to this question we performed further experiments addressing the question of antigen specificity, comparing responses of pOVA-specific OT-II cells with those of SMARTA T cells which are specific for the irrelevant LCMV GP-derived P13 peptide. Results from these experiments provided further evidence supporting the hypothesis that we were observing antigen-specific responses of OT-II cells within the glomerulus (see **Fig. 1** above). These new data have been added to an updated **Figure 2** in the revised manuscript.

1.2 It is critical to examine the activation state of OT-II cells in other organs (spleen, blood, liver, bone marrow, lymph nodes) of 8D1/pOVA and control treated animals.

Response: To address this point, we performed additional experiments examining OT-II cell activation in blood spleen, lymph nodes, liver and bone marrow, in animals treated with 8D1, 8D1/pOVA and IgG/pOVA (**Fig. 9** below). These experiments revealed that antigen-dependent OT-II cell activation was minimal in the blood, spleen and lymph nodes, suggesting that administration of 8D1/pOVA does not result in activation of T cells that were recirculating systemically. In contrast, significant T cell activation occurred in the liver in the presence of 8D1/pOVA but not when equimolar amounts of unconjugated 8D1 or pOVA conjugated to a control antibody (IgG/pOVA) were administered. This response is interesting and consistent with 8D1 specifically localising to a target antigen in the liver, but does not affect the interpretations of our findings. Finally, 20-30% of OT-II cells that homed to the bone marrow were found to be positive for IFN γ generation irrespective of the type of antibody or mode of pOVA administration, indicating that this

microenvironment was supporting antigen-independent T cell activation. Together these data show that while the kidney is not the only location where OT-II cells undergo activation following administration of 8D1/pOVA, under these conditions activated T cells are not present in the circulation (and therefore capable of migrating to the kidney) or in secondary lymphoid organs where recirculating lymphocytes are often found.

Figure 9: OT-II cell activation in organs of mice treated with glomerulus-targeted vs non-targeted pOVA. Assessment of OT-II cell IFN γ production in the blood, spleen, lymph nodes (LN), liver and bone marrow (BM) in the presence of glomerulus-targeting antibody alone (8D1), glomerulus-targeted antigenic peptide (8D1/pOVA), or non-targeted antigenic peptide (IgG/pOVA). Group data showing IFN γ expression (as % of OT-II cells) in blood, spleen, LN, liver and BM. Data are shown as mean \pm sem of n=7 (8D1) or 8 (8D1/pOVA & IgG/pOVA) mice/group. **, P<0.01.

are shown as mean \pm sem of n=7 (8D1) or 8 (8D1/pOVA & IgG/pOVA) mice/group. **, P<0.01.

2. Even if the activated OT-II cells were exclusively found in the kidney, this does not inevitably lead to the conclusion that activation occurred essentially within glomeruli. Multiple micrographs in the paper indicate that there is an abundance of MHC-II⁺ cells in extraglomerular renal tissue. How can the authors rule out that T cell activation did not occur in other parts of the kidney or by interactions with periglomerular MHC-II⁺ cells that may extend processes into the vessel lumen, as has been shown in the skin? Are peri-glomerular renal MHC-II⁺ cells susceptible to deletion by clodronate liposomes?

Response: Firstly as described in the responses above, we have now performed experiments using OT-II cells expressing an NFAT-GFP reporter construct which demonstrate that OT-II cells do undergo activation within the glomerular capillaries (see response to **Reviewer #1, comment #2** & response to **Reviewer #2, comment #2**). These experiments have resulted in substantial additions to the revised manuscript, as part of a revised **Figure 4 (panels d-g)**, **Suppl. Figure 4** and in the new **Suppl. Video S8**).

With regard to the question of whether periglomerular MHCII⁺ cells extend processes into glomerular capillaries, Stamatiades *et al.*, recently confirmed our observations regarding the absence of intraglomerular mononuclear phagocytes, or their projections, in the glomerulus (22) (see Video S2 in this paper). As such, we do not believe that extraglomerular mononuclear phagocytes inserting processes into glomerular capillaries is a likely mechanism by which effector CD4⁺ T cells recognise antigen in glomerular capillaries.

Regarding the sensitivity of periglomerular MHCII⁺ cells to clodronate, we performed additional experiments to examine the sensitivity of extraglomerular MHCII⁺ cells in the kidney to clodronate treatment. These experiments revealed that 24 h after clodronate administration, these cells were mostly eliminated (see **Fig. 10** below). These findings are consistent with the recognised capacity of clodronate liposomes to deplete tissue-resident mononuclear phagocytes and indicate that in experiments in which we used clodronate to remove monocytes, intrarenal mononuclear phagocytes were also eliminated. This result, while not surprising, does not invoke a role for these cells in antigen recognition in this model.

Figure 10: Effect of clodronate on extraglomerular MHCII⁺ cells in the kidney. MHCII-GFP mice were treated with either control liposomes (Control) or clodronate-containing liposomes (Clodronate) and the region occupied by MHCII⁺ cells was determined (expressed as a % area of optical sections in multiphoton images of fixed kidneys). N=3 for each treatment, data shown as mean \pm sem.

3. Intravital imaging studies of CD4 T cells in other tissues indicates that CD4 T cells, unlike CD8 T cells, require prolonged stable interactions with APCs (for several hours) to achieve an activated state. According to Fig. 4b and c, intravascular OT-II cells in glomeruli interacted, on average, with only one MHC-II⁺ cell for less than 5 minutes, and all contacts lasted less than 50 min. It is unclear whether such short contacts could be productive. It is conceivable that the *ex vivo* conditioning of OT-II cells prior to adoptive transfer may have lowered the requirement of T cells for extended antigen exposure, but this would have to be properly documented. Moreover, it needs to be shown that the observed T cell response is not merely an artifact of the non-physiological *ex vivo* conditioning protocol. Data with endogenously generated effector/memory cells need to be included.

Response: Published studies have repeatedly demonstrated that effector CD4⁺ T cells in peripheral tissues can very rapidly recognise and respond to antigen. The studies to which the reviewer refers examine naïve T cells undergoing antigen-induced activation in lymph nodes. It is important to recognise that the T cells used in our experiments have already acquired the capacity to act as effectors when they recognise antigen, and are thus effector cells when they arrive at the glomerulus. As such, the response we are modelling in the glomerulus is the recognition of antigen in the periphery, by antigen-experienced effector T cells. As an indication of the speed with which these cells can respond, the work of Ron Germain's laboratory clearly demonstrates that effector T cells, localised to the periphery, can respond almost instantaneously to antigen administration and presentation, as detected by assessment of reduced motility (4). In addition, within 2 hrs, these effector cells can robustly produce IFN γ , detectable via intracellular staining (4, 12). Thus the time course of antigen-induced responses of effector T cells, as examined in the present study, is quite different from that of naïve T cells undergoing antigen-induced activation in lymph nodes. Our new studies using the NFAT-GFP reporter system *in vivo* are also consistent with rapid signalling by effector T cells upon antigen recognition in the glomerulus. To emphasise and clarify this point, we have added the following paragraph to the Discussion:

"Alterations in migration are characteristic of T cells undergoing antigen recognition. For naïve cells, altered migration takes several hours to become apparent.(23, 24) In contrast, effector T cells can respond extremely rapidly to antigen presentation, as evidenced by rapid migratory arrest and induction of IFN γ production.(4, 12) The T cell behavior observed in the present study was consistent with previous descriptions of effector T cells. Average migration velocity of all OT-II cells was decreased within the first two hours of the response. More tellingly, identification of cells undergoing antigen presentation on the basis of nuclear localization of NFAT-GFP revealed that these cells were uniformly static, while cells with non-translocated NFAT-GFP, and therefore not detecting antigen, continued to migrate. This T cell response occurred within the first hour after T cell transfer. Analysis of cytokine production provided further evidence of rapid antigen-specific activation by intrarenal T cells in that many intrarenal effector CD4⁺ T cells were positive for IFN γ production 4 h after transfer. Finally, changes in neutrophil behavior resulting from this rapid T cell response were apparent as early as four hours after initiation. These findings highlight the speed with

which effector T cells can respond to antigen in the periphery and promote downstream inflammation.”

4. No evidence is presented that the small fraction of MHC-II⁺ monocytes in peripheral blood are capable of capturing, processing and presenting antigen to CD4 T cells. It would be necessary to sort MHC-II⁺ monocytes from animals injected with pOVA/8D1 and perform an *in vitro* Ag presentation assay without any additional priming. Which costimulatory molecules and cytokines are involved in this process?

Response: As discussed in the response to Reviewer 1 (Comment #3), to address this point, we sorted CD115⁺ MHCII⁺ monocytes from the blood of MHCII-GFP mice (excluding CD19⁺ B cells, which constitute ~95% of the MHCII⁺ cells in the circulation), and used this population in OT-II cell proliferation assays, as we have done with other APC populations in the past (15). These experiments revealed that MHCII⁺ monocytes have the capacity to induce OT-II T cell proliferation under conditions where antigenic OVA-peptide is present (see **Fig. 6** above). In these experiments, the level of T cell proliferation was proportional to the number of monocytes in the assay and T cells did not proliferate in the absence of APCs, supporting the specificity of this response. These observations indicate that the MHCII-expressing subset of circulating monocytes does have the capacity to present antigen to CD4⁺ T cells and induce their proliferation. These new data, have been added to **Figure 7** in the revised manuscript.

5. It is known that the CX3CR1⁺ patrolling monocytes express MHC-II and, in fact, previous studies have already reported a role for these monocytes in neutrophil retention and glomerulonephritis. The authors should check the expression of CX3CR1 in the MHC-II⁺ CD11b⁺ cells studied herein, since they might be the same population as previously studied.

Response: It should be noted that, as we show in this study and as others have observed, not all patrolling monocytes express MHCII – only a small proportion of them do (e.g. see **Figure 6** in the revised manuscript showing ~ 16% of these cells express MHCII). To address the reviewer’s comment, we performed additional flow cytometry experiments to examine expression of CX3CR1 in the MHCII⁺ monocyte population. Monocytes could be divided into separate populations of CX3CR1^{high} and CX3CR1^{low-intermediate} expression (**Fig. 11a**, below), as previously described (25). Consistent with these previous observations, the CX3CR1^{high} population comprised the Ly6C^{lo/neg} non-classical subset, while the CX3CR1^{low-intermediate} population comprised the Ly6C⁺ classical monocyte subset (**Fig. 11a**, below). MHCII⁺ cells were present at low abundance in each of these subsets, typically representing ~ 10% of the total monocyte population (**Fig. 11b**, below). In summary, these findings demonstrate that while monocytes express CX3CR1 to varying degrees, only a subset of these cells express MHCII, with these cells being present in both the CX3CR1^{high} non-classical ‘patrolling’ subset and the CX3CR1^{low-intermediate} classical subset. These new data are now included in the revised manuscript as **Supplementary Figure 6**.

Figure 11. Flow cytometric assessment of monocyte expression of CX3CR1 and MHCII. (a) Single mononuclear blood leukocytes were assessed for expression of CX3CR1

and Ly6C, identifying distinct CX3CR1^{hi} Ly6C^{-ve} and CX3CR1^{lo-int} Ly6C⁺ monocyte populations. (b) Monocytes (identified on the basis of CD115 expression) were assessed for expression of CX3CR1 and MHCII. Shown are % of monocytes in each of the following gates: CX3CR1^{hi} MHCII^{-ve}, CX3CR1^{hi} MHCII⁺, CX3CR1^{lo-int} MHCII^{-ve}, and CX3CR1^{lo-int} MHCII⁺. Data show a representative example of n=3 mice, from two individual experiments.

6. Clodronate liposomes are a blunt tool to address the questions at hand. Systemic treatment with this reagent may not only deplete monocytes, but macrophages and dendritic cells in both intra- and (some) extravascular compartments. The effects observed with this treatment may not necessarily be attributed to monocytes. How does clodronate impact the adhesion and crawling of CD4 effector T cells?

Response: While we agree that other mononuclear phagocytes are affected by clodronate, multiple experiments presented in the revised manuscript implicate intravascular monocytes in intravascular antigen recognition in the glomerulus. To exclude a direct effect of clodronate on the responding CD4⁺ T cells, we performed additional experiments examining the behaviour of effector OT-II CD4⁺ T cells in glomeruli (without antigen) of mice pre-treated with clodronate or control liposomes. In mice treated with clodronate liposomes, adhesion and dwell time of effector T cells within the glomerulus were not significantly different from those in mice receiving control liposomes (**Fig. 12** below). These experiments provide evidence that clodronate treatment alone was not sufficient to induce changes in T cell behaviour. These observations have now been described (as **data not shown**) in the revised manuscript.

Figure 12: Clodronate treatment does not alter effector T cell behaviour in glomeruli. Mice were treated with either control or clodronate-loaded liposomes and subsequently injected with 1×10^7 activated OT-II T cells. T cell behavior in glomeruli was assessed by multiphoton microscopy in the 2 hours following injection. Data are shown for number (A) and dwell time (B) of adherent cells. Data are shown as mean \pm sem of n=5 mice/group.

7. To rule out the possibility that glomerulonephritis in 8D1/pOVA treated mice was not dependent on T cell activation, authors need to show that control animals injected with unconjugated 8D1 and OT-II cells do not develop glomerulonephritis. Based on the prior literature, the injection of the antibody per se can be sufficient to produce monocyte-induced retention of neutrophils within glomeruli.

Response: Multiple pieces of evidence demonstrate that 8D1 at the dose used in this study is non-nephritogenic, including in the presence of activated OT-II cells:

1. In the paper referred to regarding monocyte-induced retention of neutrophils within glomeruli, these responses were induced by injection of 15 mg of anti-glomerular basement antibody. In the present manuscript, mice are injected with 150 μ g of 8D1, markedly reducing the likelihood of inducing antibody-dependent effects.

2. In our original description of this model, we showed that 8D1/pOVA, administered in the absence of OT-II cells, did not induce histopathological glomerulonephritis or albuminuria when examined 7-21 days later (26).
3. We previously demonstrated that mice that receiving 150 µg 8D1 alone did not undergo acute (4 h) neutrophil recruitment (6). Similarly, mice that received activated OT-II cells, as performed in the present study, along with 150 µg of 8D1 conjugated to an irrelevant peptide, did not develop glomerulonephritis (6).
4. In our previous multiphoton study examining leukocyte responses in the glomerulus, we demonstrated that administration of unconjugated 8D1 and activated OT-II cells failed to induce alterations in retention of neutrophils and monocytes above control levels, when examined at 4, 24 and 48 h after initiation of the model (Devi *et al.*, **Suppl. Fig. 7**) (15). Consistent with this, in the present manuscript, OT-II cells failed to produce IFN γ in mice receiving unconjugated 8D1 and activated OT-II cells. Similarly, intraglomerular neutrophils showed minimal signs of activation in these animals.

Together these observations indicate that that 8D1 at the dose used in this study does not induce glomerulonephritis.

REFERENCES

1. Pesic, M., I. Bartholomaeus, N. I. Kyrtasous, V. Heissmeyer, H. Wekerle, and N. Kawakami. 2013. 2-photon imaging of phagocyte-mediated T cell activation in the CNS. *J. Clin. Invest.* 123: 1192-1201.
2. Marangoni, F., T. T. Murooka, T. Manzo, E. Y. Kim, E. Carrizosa, N. M. Elpek, and T. R. Mempel. 2013. The transcription factor NFAT exhibits signal memory during serial T cell interactions with antigen-presenting cells. *Immunity* 38: 237-249.
3. Macian, F. 2005. NFAT proteins: key regulators of T-cell development and function. *Nat. Rev. Immunol.* 5: 472-484.
4. Honda, T., J. G. Egen, T. Lämmermann, W. Kastenmüller, P. Torabi-Parizi, and R. N. Germain. 2014. Tuning of Antigen Sensitivity by T Cell Receptor-Dependent Negative Feedback Controls T Cell Effector Function in Inflamed Tissues. *Immunity* 40: 235-247.
5. Hutton, H. L., S. R. Holdsworth, and A. R. Kitching. 2017. ANCA-Associated Vasculitis: Pathogenesis, Models, and Preclinical Testing. *Semin. Nephrol.* 37: 418-435.
6. Ooi, J. D., J. Chang, M. J. Hickey, D. B. Borza, L. Fugger, S. R. Holdsworth, and A. R. Kitching. 2012. The immunodominant myeloperoxidase T-cell epitope induces local cell-mediated injury in antimyeloperoxidase glomerulonephritis. *Proc. Natl. Acad. Sci. U. S. A.* 109: E2615-2624.
7. Brouwer, E., C. A. Stegeman, M. G. Huitema, P. C. Limburg, and C. G. Kallenberg. 1994. T cell reactivity to proteinase 3 and myeloperoxidase in patients with Wegener's granulomatosis (WG). *Clin. Exp. Immunol.* 98: 448-453.
8. Cairns, L. S., R. G. Phelps, L. Bowie, A. M. Hall, W. W. Saweirs, A. J. Rees, and R. N. Barker. 2003. The fine specificity and cytokine profile of T-helper cells responsive to the alpha3 chain of type IV collagen in Goodpasture's disease. *J. Am. Soc. Nephrol.* 14: 2801-2812.
9. Popa, E. R., C. F. Franssen, P. C. Limburg, M. G. Huitema, C. G. Kallenberg, and J. W. Tervaert. 2002. In vitro cytokine production and proliferation of T cells from patients with anti-proteinase 3- and antimyeloperoxidase-associated vasculitis, in response to proteinase 3 and myeloperoxidase. *Arthritis Rheum.* 46: 1894-1904.
10. Monti, P., M. Scirpoli, A. Rigamonti, A. Mayr, A. Jaeger, R. Bonfanti, G. Chiumello, A. G. Ziegler, and E. Bonifacio. 2007. Evidence for in vivo primed and expanded autoreactive T cells as a specific feature of patients with type 1 diabetes. *J. Immunol.* 179: 5785-5792.
11. Scally, S. W., J. Petersen, S. C. Law, N. L. Dudek, H. J. Nel, K. L. Loh, L. C. Wijeyewickrema, S. B. Eckle, J. van Heemst, R. N. Pike, J. McCluskey, R. E. Toes, N. L. La Gruta, A. W. Purcell, H. H. Reid, R. Thomas, and J. Rossjohn. 2013. A molecular basis for the association of the HLA-DRB1 locus, citrullination, and rheumatoid arthritis. *J. Exp. Med.* 210: 2569-2582.
12. Egen, J. G., A. Rothfuchs, C. G. Feng, M. A. Horwitz, A. Sher, and R. N. Germain. 2011. Intravital Imaging Reveals Limited Antigen Presentation and T Cell Effector Function in Mycobacterial Granulomas. *Immunity* 34: 807-819.
13. Snelgrove, S. L., J. Y. Kausman, C. Lo, C. Lo, J. D. Ooi, P. T. Coates, M. J. Hickey, S. R. Holdsworth, C. Kurts, D. R. Engel, and A. R. Kitching. 2012. Renal dendritic cells adopt a pro-inflammatory phenotype in obstructive uropathy to activate T cells but do not directly contribute to fibrosis. *Am. J. Pathol.* 180: 91-103.
14. Snelgrove, S. L., C. Lo, P. Hall, C. Y. Lo, M. A. Alikhan, P. T. Coates, S. R. Holdsworth, M. J. Hickey, and A. R. Kitching. 2017. Activated renal dendritic cells cross present intrarenal antigens after ischemia reperfusion injury. *Transplantation* 105: 1013-1024.
15. Devi, S., A. Li, C. L. Westhorpe, C. Y. Lo, L. D. Abeynaïke, S. L. Snelgrove, P. Hall, J. D. Ooi, C. G. Sobey, A. R. Kitching, and M. J. Hickey. 2013. Multiphoton imaging reveals a new leukocyte recruitment paradigm in the glomerulus. *Nat. Med.* 19: 107-112.
16. Finsterbusch, M., P. Hall, A. Li, S. Devi, C. L. V. Westhorpe, A. R. Kitching, and M. J. Hickey. 2016. Patrolling monocytes promote intravascular neutrophil activation and glomerular injury in the acutely-inflamed glomerulus. *Proc Natl Acad Sci U S A* 113: E5172-E5181.
17. Hwang, J. M., J. Yamanouchi, P. Santamaria, and P. Kubes. 2004. A critical temporal window for selectin-dependent CD4+ lymphocyte homing and initiation of late-phase inflammation in contact sensitivity. *J. Exp. Med.* 199: 1223-1234.

18. Looney, M. R., E. E. Thornton, D. Sen, W. J. Lamm, R. W. Glenny, and M. F. Krummel. 2011. Stabilized imaging of immune surveillance in the mouse lung. *Nature methods* 8: 91-96.
19. Thanabalasuriar, A., B. G. Surewaard, M. E. Willson, A. S. Neupane, C. K. Stover, P. Warrener, G. Wilson, A. E. Keller, B. R. Sellman, A. DiGiandomenico, and P. Kubes. 2017. Bispecific antibody targets multiple *Pseudomonas aeruginosa* evasion mechanisms in the lung vasculature. *J Clin Invest* 127: 2249-2261.
20. Yipp, B. G., J. H. Kim, R. Lima, L. D. Zbytniuk, B. Petri, N. Swanlund, M. Ho, V. G. Szeto, T. Tak, L. Koenderman, P. Pickkers, A. T. J. Tool, T. W. Kuijpers, T. K. van den Berg, M. R. Looney, M. F. Krummel, and P. Kubes. 2017. The Lung is a Host Defense Niche for Immediate Neutrophil-Mediated Vascular Protection. *Sci Immunol* 2: pii: eaam8929.
21. Devi, S., M. P. Kuligowski, R. Y. Kwan, E. Westein, S. P. Jackson, A. R. Kitching, and M. J. Hickey. 2010. Platelet recruitment to the inflamed glomerulus occurs via an α IIb β 3/GPVI-dependent pathway. *Am J Pathol* 177: 1131-1142.
22. Stamatiades, E. G., M. E. Tremblay, M. Bohm, L. Crozet, K. Bisht, D. Kao, C. Coelho, X. Fan, W. T. Yewdell, A. Davidson, P. S. Heeger, S. Diebold, F. Nimmerjahn, and F. Geissmann. 2016. Immune Monitoring of Trans-endothelial Transport by Kidney-Resident Macrophages. *Cell* 166: 991-1003.
23. Mempel, T. R., S. E. Henrickson, and U. H. von Andrian. 2004. T-cell priming by dendritic cells in lymph nodes occurs in three distinct phases. *Nature* 427: 154-159.
24. Miller, M. J., S. H. Wei, I. Parker, and M. D. Cahalan. 2002. Two-photon imaging of lymphocyte motility and antigen response in intact lymph node. *Science* 296: 1869-1873.
25. Auffray, C., D. Fogg, M. Garfa, G. Elain, O. Join-Lambert, S. Kayal, S. Sarnacki, A. Cumano, G. Lauvau, and F. Geissmann. 2007. Monitoring of blood vessels and tissues by a population of monocytes with patrolling behavior. *Science* 317: 666-670.
26. Summers, S. A., S. M. Steinmetz, M. Li, J. Y. Kausman, T. J. Semple, K. L. Edgton, D. B. Borza, H. Braley, S. R. Holdsworth, and A. R. Kitching. 2009. Th1 and Th17 cells induce proliferative glomerulonephritis. *J. Am. Soc. Nephrol.* 20: 2518-2524.

REVIEWERS' COMMENTS:

Reviewer #1 (Remarks to the Author):

The authors have done an outstanding job of revising the manuscript to address the previous criticisms and my recommendation is to accept it for publication in Nature Communications. The authors added substantial new data to strengthen the conclusions, clarify various technical issues and enhance the overall significance of the study. Honestly, I don't remember ever seeing a more thorough, thoughtful and convincing rebuttal to a manuscript critique!

Reviewer #2 (Remarks to the Author):

My only real concern about the original version of the manuscript by Westhorpe et al was whether the data presented proved that the behavior of OTII cells within the glomerulus after injection of 8D1/pOVA was due to local presentation of antigen by patrolling monocytes or left open the possibility that presentation occurred elsewhere.

The data from three new sets of experiments presented in the revised manuscript eliminate this possibility. These data convincingly demonstrate: (i) IgG/pOVA does not affect OTII cell behavior in glomerular capillaries as 8D1/pOVA does (the experiment I had suggested); (ii) the behavior of adoptively transferred TCR transgenic T cells with specificity for an irrelevant (CMV) peptide is not affected by injection of 8D1/pOVA; and most elegantly (iii) by demonstrating the specific effect 8D1/OVA on cytoplasmic to nuclear translocation of NFAT within glomerular capillaries (and the altered behavior of OTII with nuclear NFAT). These data unequivocally prove the authors hypothesis that monocytes patrolling glomerular capillaries present locally acquired antigen to memory T cells in vivo. This is highly novel and potentially very important.

The authors are right to correct my false assumption that the T whole kidney T cell experiments were performed on hydronephrotic mice in my second comment; and their NFAT experiments convincingly address the substantive point.

My third and fourth comments are both are well addressed by the changes to the text - the transmission EM picture is a bonus!

Finally, the authors should be congratulated on the rewritten results section which is a model of how to present complex data with concision and clarity.

Reviewer #3 (Remarks to the Author):

The authors clearly have made an effort to address most of the concerns raised by the reviewers. Unfortunately, there are still a few key points that remain unanswered despite the new experimentation. Specifically, the authors may consider the following:

1. This reviewer's previous critique had pointed out that the effector T cells that were used throughout the study were generated following a non-physiological ex vivo conditioning protocol. The question whether the model can also be demonstrated to apply to endogenous effector/memory cells remains unanswered.

2. A key message of this paper is the claim that MHC-II+ monocytes can capture, process and present antigen to CD4+ T cells intravascularly. However, this is still not fully established in this revised version of the manuscript. In an attempt to demonstrate the APC function for MHC-II+ monocytes, the authors have performed an in vitro co-culture assay using OVA-specific T cells and

MHC-II+ GFP monocytes that were pulsed with OVA peptide in vitro. This experiment is far from sufficient to demonstrate a role for MHC-II+ monocytes as intravascular APCs in vivo, since it obviates the need to capture and process antigenic protein and efficiently present protein-derived peptides in MHC-II. As previously mentioned, a more definitive strategy to address this critical point would be the demonstration that MHC-II+ monocytes isolated from peripheral blood or kidneys of animals injected with pOVA/8D1 can stimulate OVA-specific CD4+ T cells without requiring additional in vitro antigen pulsing.

Minor comment

Regarding the expression of CX3CR1 in MHC-II+ monocytes, the interpretation of the FACS data is somewhat puzzling. In the FACS blots the MHC-II+ monocytes appear as a homogenous population expressing intermediate levels of CX3CR1. However, the authors state that ~35% of MHC-II+ monocytes were CX3CR1-high (patrolling subset), whereas ~65% were CX3CR1-int/low (classical subset). What was the level of expression of Ly6C in MHC-II+ monocytes? Without showing Ly6C in conjunction with CX3CR1, it is difficult to determine with confidence whether MHC-II+ monocytes belong to the classical or the patrolling subset (or both).

NCOMMS-16-28698B - Rebuttal

Effector CD4⁺ T cells recognize intravascular antigen presented by patrolling monocytes

In order to address the remaining issues raised by Reviewer 3, we have made the following modifications to the manuscript:

1. The question whether the model can also be demonstrated to apply to endogenous effector/memory cells remains unanswered.

RESPONSE: To address this point, we have added the following sentence to the Discussion:

“In these experiments, we used effector T cells generated via a standard *ex vivo* differentiation protocol. It is conceivable that responses of cells generated in this manner may differ from those of cells differentiated *in vivo*, a possibility that could be explored in future studies.”

*2. A key message of this paper is the claim that MHC-II+ monocytes can capture, process and present antigen to CD4+ T cells intravascularly. However, this is still not fully established in this revised version of the manuscript... A more definitive strategy to address this critical point would be the demonstration that MHC-II+ monocytes isolated from peripheral blood or kidneys of animals injected with pOVA/8D1 can stimulate OVA-specific CD4+ T cells without requiring additional *in vitro* antigen pulsing.*

RESPONSE: To address this point, we have added the following sentence to the Discussion:

“We also demonstrated that MHCII+ monocytes can induce antigen-specific T cell proliferation *in vitro*, providing additional support for the contention that these cells are responsible for antigen-specific T cell activation in glomerular capillaries. With future technical advances, it may be feasible to isolate or image patrolling MHCII+ monocytes in glomeruli to demonstrate antigen uptake from within the glomerular microvasculature.”